# MiDAS 5: Global diversity of bacteria and archaea in anaerobic digesters

Morten Kam Dahl Dueholm [1] ✉, Kasper Skytte Andersen [1], Anne-Kirstine C. Korntved[1], Vibeke Rudkjøbing[1], Madalena Alves[2], Yadira Bajón-Fernández [3], Damien Batstone[4], Caitlyn Butler[5], Mercedes Cecilia Cruz[6], Åsa Davidsson [7], Leonardo Erijman[8], Christof Holliger [9], Konrad Koch [10], Norbert Kreuzinger[11], Changsoo Lee[12], Gerasimos Lyberatos [13], Srikanth Mutnuri[14], Vincent O'Flaherty[15], Piotr Oleskowicz-Popiel[16], Dana Pokorna[17], Veronica Rajal[6], Michael Recktenwald[18], Jorge Rodríguez [19], Pascal E. Saikaly [20], Nick Tooker[5], Julia Vierheilig [11], Jo De Vrieze [21], Christian Wurzbacher[10] & Per Halkjær Nielsen [1] ✉

Anaerobic digestion of organic waste into methane and carbon dioxide (biogas) is carried out by complex microbial communities. Here, we use full-length 16S rRNA gene sequencing of 285 full-scale anaerobic digesters (ADs) to expand our knowledge about diversity and function of the bacteria and archaea in ADs worldwide. The sequences are processed into full-length 16S rRNA amplicon sequence variants (FL-ASVs) and are used to expand the MiDAS 4 database for bacteria and archaea in wastewater treatment systems, creating MiDAS 5. The expansion of the MiDAS database increases the coverage for bacteria and archaea in ADs worldwide, leading to improved genus- and species-level classification. Using MiDAS 5, we carry out an amplicon-based, global-scale microbial community profiling of the sampled ADs using three common sets of primers targeting different regions of the 16S rRNA gene in bacteria and/or archaea. We reveal how environmental conditions and biogeography shape the AD microbiota. We also identify core and conditionally rare or abundant taxa, encompassing 692 genera and 1013 species. These represent 84–99% and 18–61% of the accumulated read abundance, respectively, across samples depending on the amplicon primers used. Finally, we examine the global diversity of functional groups with known importance for the anaerobic digestion process.

Anaerobic digestion has gained attention as an important, sustainable biotechnology as it provides several benefits that align with the goals of sustainability. It can help to produce renewable energy (biogas) from organic waste such as manure, food waste, and sludge from wastewater treatment plants (WWTPs)[1,2]. The anaerobic digestion process also reduces pathogens and the amount of organic waste that is sent to landfills, thereby reducing methane emissions and supporting sustainable waste management practices[1]. Finally, the fertilizer that is produced as a byproduct of anaerobic digestion can be used to support sustainable agriculture, reducing the need for synthetic fertilizers that can have negative environmental impacts[3,4].

The anaerobic digestion process relies on the microbial degradation and conversion of organic matter, which requires a complex interplay between several functional guilds. These include hydrolyzing, acidogenic, and acetogenic syntrophic bacteria as well as methanogenic archaea[5]. The taxonomy is poorly characterized for many of the microorganisms in anaerobic digesters (ADs), and even among the most abundant taxa many lack genus- or species-level classifications[6]. To optimize performance, a comprehensive knowledge about microbial immigration/competition, environmental/operational conditions, and taxonomy is essential[7–9]. Recent microbial surveys have increased our knowledge about the anaerobic digestion process[7,10–16]. However, sharing knowledge across studies is still hindered by the absence of standardized protocols and a common reference database with a unifying taxonomy[17,18]. To facilitate collaboration and knowledge sharing, it is essential to establish these standard protocols and resources.

The Microbial Database for Activated Sludge and Anaerobic Digesters (MiDAS) project was established as an open-source platform for sharing updated knowledge about the physiology and ecology of the important microorganisms present in engineered ecosystems of activated sludge plants, ADs, and related WWTPs[17–20]. MiDAS provides standardized protocols for microbial profiling of microbes in wastewater treatment systems[21], an ecosystem-specific full-length 16S rRNA gene reference database[18,20], and a field guide where knowledge about the specific genera are stored and shared (https://www.midasfieldguide.org).

The MiDAS 16S rRNA gene reference database was created based on millions of high-quality, chimera-free, full-length 16S rRNA genes resolved into amplicon sequence variants (ASVs) and classified using automated taxonomy assignment (AutoTax)[6,18,20].

AutoTax provides a comprehensive seven-rank taxonomy (kingdom to species-level) for all reference sequences based on the most recent version of the SILVA SSURef 99 NR taxonomy and includes a robust placeholder taxonomy for lineages without an official taxonomy[6]. The placeholder taxa are easily distinguishable by their names, formatted as 'midas_x_y', where 'x' indicates the taxonomic rank and 'y' is a numerical identifier. This naming convention facilitates the study of unclassified alongside classified taxa across various taxonomic ranks. The placeholder taxonomy should not be seen as a replacement for proper taxonomic classifications but can

pinpoint important lineages that should be studied in depth using phylogenomics[22–26].

The MiDAS 16S rRNA gene reference database (MiDAS 4.8.1) currently contains reference sequences from WWTPs worldwide and ADs located at WWTPs in Denmark[20]. However, it may not provide comprehensive coverage for all important microbes found in ADs treating other types of waste or in other locations.

In this study, we introduce MiDAS 5, an updated version of MiDAS 4 expanded with more than half a million high-quality, full-length archaeal and bacterial 16S rRNA gene sequences from 285 ADs worldwide treating different types of biowaste. We carried out a global survey of ADs using three commonly used short-read amplicon primer sets targeting bacteria (V1-V3), archaea (V3-V5), and both (V4). This data was used in combination with MiDAS 5 to (i) link the global diversity of bacteria and archaea to biogeography and environmental factors, (ii) identify important core taxa, and (iii) uncover the global diversity within selected functional guilds. The results provide a solid foundation for future research on AD microbiology.

## Results and Discussion

The MiDAS Global Consortium for Anaerobic Digesters was established in 2018 to coordinate the sampling and collection of metadata from ADs worldwide (Supplementary Data 1). Samples were obtained in duplicates from 285 ADs in 196 cities in 19 countries on five continents (Fig. 1a). Most of the ADs treated surplus sludge from WWTPs (69.8%) (Fig. 1b). However, ADs treating food waste (8.1%), industrial waste (7.4%), and manure (5.3%) were also included in the survey. Most of the ADs were mesophilic (86.0%), few were thermophilic (6.0%), and the rest did not provide temperature data (8.1%). The main digester technology used was continuous stirred-tank reactors (67.7%) followed by two-stage reactors (12.6%). A few upflow anaerobic sludge blanket (UASB) and other types were also sampled to expand the diversity of digester types.

### Expanding the MiDAS database with reference sequences from global ADs

To expand the MiDAS database with sequences from ADs across the globe, we applied high-fidelity, full-length 16S rRNA gene sequencing on all samples collected in this study. More than half a million full-length 16S rRNA gene sequence reads, representing both bacteria

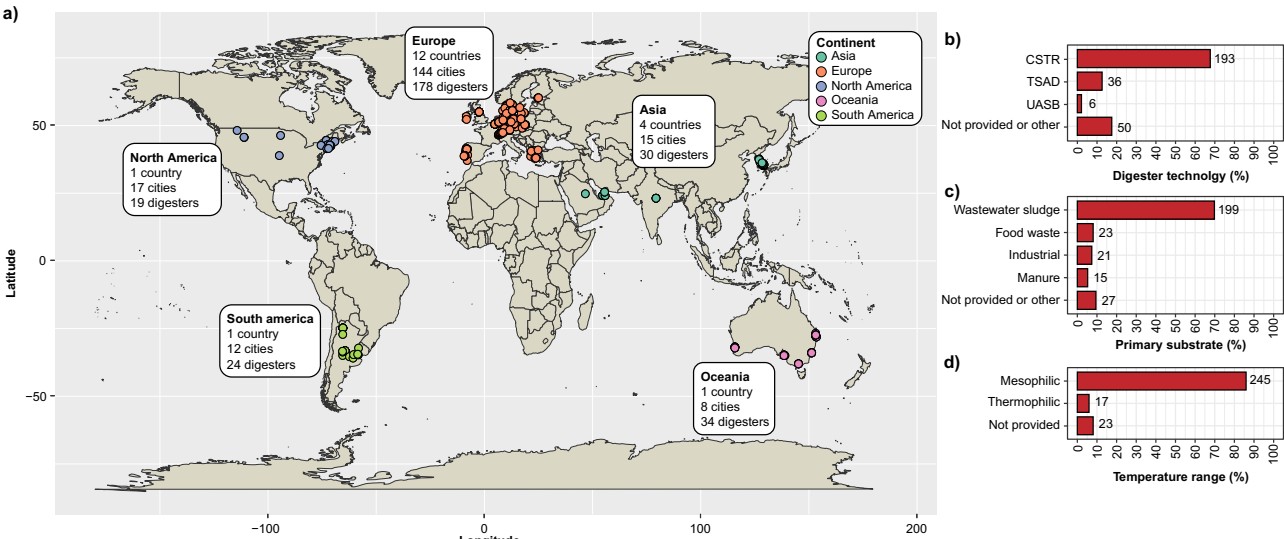

**Fig. 1 | Sampling of anaerobic digesters (ADs) across the world. a** Geographical distribution of ADs included. **b** Distribution of digester technologies. CSTR Continuous stirred-tank reactor; TSAD Two-stage anaerobic digestion, UASB Upflow anaerobic sludge blanket. **c** Distribution of primary substrates. **d** Distribution of digester temperatures. The values next to the bars are the number of ADs in each group.

**Table 1 | Sequence novelty of FL-ASVs obtained in this study**

| | SILVA 138.1 SSURef NR99 | | MiDAS 4.8.1 | |
|---|---|---|---|---|
| | Sequences | Percentage | Sequences | Percentage |
| Phylum (<75.0%) | 28 | 0.09% | 67 | 0.22% |
| Class (<78.5%) | 55 | 0.18% | 186 | 0.61% |
| Order (<82.0%) | 112 | 0.37% | 462 | 1.53% |
| Family (<86.5%) | 354 | 1.17% | 1483 | 4.90% |
| Genus (<94.5%) | 5240 | 17.32% | 9419 | 31.14% |
| Species (<98.7%) | 15,806 | 52.26% | 16,863 | 55.75% |

Sequence novelty was determined based on the percentage identity between each of the 30,246 new FL-ASVs and their closest relative in the databases indicated and identity thresholds for each taxonomic rank proposed by Yarza et al.[27] shown in the parentheses.

**Table 2 | New taxa introduced with MiDAS 5**

| | Total taxa | New taxa | Increase (%) |
|---|---|---|---|
| Phylum | 105 | 20 | 23.5% |
| Class | 259 | 37 | 16.7% |
| Order | 727 | 104 | 16.7% |
| Family | 2212 | 360 | 19.4% |
| Genus | 12,254 | 2770 | 29.2% |
| Species | 40,207 | 8858 | 28.3% |

The number of new taxa represent unique taxa at the different taxonomic ranks that were not part of MiDAS 4.8.1[20] and includes both official taxonomic names and de novo placeholder names provided by AutoTax[6].

and archaea, were obtained after quality filtering and primer trimming. After processing the sequence reads with AutoTax to produce full-length 16S rRNA gene ASVs (FL-ASVs), these were compared and added to the existing 90,164 FL-ASVs in the MiDAS 4.8.1 database. The combined number was then deduplicated, resulting in a total of 120,408 non-redundant FL-ASVs in the expanded MiDAS 5 database. This represents an increase of 30,246 new FL-ASVs when compared to the previous version.

The novelty of the 30,246 new FL-ASVs was determined based on the percent identity shared with their closest relatives in the SILVA 138.1 SSURef NR99 and MiDAS 4.8.1 database using the threshold for each taxonomic rank proposed by Yarza et al.[27] (Table 1). It should be noted that these thresholds do not uniformly apply across the bacterial phylogenetic tree; therefore, our taxonomic assignments should be considered as approximations intended to facilitate biological interpretation. 17% and 31% of the new FL-ASVs lacked genus-level homologs (≥94.5% identity) and 52% and 56% were without species-level homologs (≥98.7% identity) in SILVA 138.1 and MiDAS 4, respectively. This suggests a substantial increase in the diversity within the MiDAS 5 database.

**MiDAS 5 introduces many new taxa**
To investigate how the new FL-ASVs affected the taxonomic diversity in the MiDAS database, we determined the number of additional taxa introduced at different taxonomic ranks (Table 2). A substantial increase in diversity was observed with the addition of 2770 new genera (29.2% increase) and 8858 new species (28.3% increase). However, many additional taxa were also introduced at higher taxonomic ranks including six more bacterial and five more archaeal phyla previously known from the SILVA taxonomy. In addition, we identified nine lineages classified as MiDAS placeholder phyla. However, phylogenetic analysis revealed that these lineages branch closely to mitochondrial sequences, indicating they are likely mitochondrial in origin. The largest percentage of the new FL-ASVs (42.8%) was found within the Firmicutes (Supplementary Fig. 1a). Firmicutes often occur in high

abundance in ADs, where they are involved in fermentation and thereby directly stimulate biogas yields[7,10,13,15,28]. A closer look into the expanded diversity within the Firmicutes revealed that new FL-ASVs were associated with several families (Supplementary Fig. 1b), including Hungateiclostridiaceae (1324 FL-ASVs), Lachnospiraceae (788 FL-ASVs), Peptostreptococcales-Tissierellales Family_XI (763 FL-ASVs), Christensenellaceae (754 FL-ASVs), Caldicoprobacteraceae (620 FL-ASVs), and Syntrophomonadaceae (555 FL-ASVs). The Syntrophomonadaceae is of special relevance, as this family includes several syntrophic fatty acid degrading bacteria, which are often the metabolic bottleneck in the overall AD process[29,30].

**MiDAS 5 provides improved coverage and classifications for AD microbiota**
The performance of the MiDAS 5 database was evaluated based on three ASV-resolved, short-read, 16S rRNA gene amplicon datasets generated from the AD samples collected in this study (Fig. 2). The V1-V3 amplicons include only bacteria and provide high phylogenetic resolution. However, the primers targeting this region have a lower coverage for the known bacterial diversity according to in silico evaluations[6,31]. The V4 amplicons include both bacteria and archaeal lineages and are commonly used due to a very good coverage of the known bacterial diversity. However, the amplicons have a weaker phylogenetic resolution compared to V1-V3, which in many cases prevent species-level classifications[6,31]. The V3-V5 amplicons cover mainly archaea and have previously been used to describe their diversity in ADs[7,10].

Our initial analysis involved non-heuristic mapping of short-read ASVs against MiDAS 5 and other widely used reference databases, including the newly released GreenGenes2[32]. This step allowed us to establish the percent identity between each ASV and its closest match across the databases. We then calculated the percentage of ASVs that have high-identity matches (≥99% identity) in each sample and database. To focus on active microbial populations, we excluded ASVs representing the rare biosphere (those with <0.01% relative abundance), which are often enriched in non-growing organisms and environmental DNA[7,10]. MiDAS 5 performed exceptionally well for bacteria with high-identity hits of 94.8% ± 4.2% (mean ± SD) for V1-V3 and 96.3% ± 2.1% for V4 ASVs, compared to 67.9% ± 19.7% and 71.4% ± 16.1% for MiDAS 4, and 61.1% ± 9.2% and 77.1% ± 7.8% for SILVA v.138.1 (Fig. 2). The complete GreenGenes2 database displayed a coverage close to that of MiDAS 5 for V4 ASVs (95.4% ± 3.3%) but a much lower coverage for V1-V3 (32.1% ± 8.9%). The reason is that the complete GreenGenes2 database contains V4 ASVs from Qiita[33] in addition to full-length 16S rRNA gene sequences[32]. For the V3-V5 archaeal dataset, an increase in coverage was observed from 33.5% ± 7.0% with MiDAS 4 to 55.9% ± 9.5% with MiDAS 5. However, the SILVA database (67.0% ± 11.0%) and the complete GTDB database (69.2% ± 13.0%) provide even better coverage. The lower coverage for archaea compared to bacteria in MiDAS 5 is likely due to reduced sequencing efforts and the challenges in designing effective universal primers for archaeal full-length 16S rRNA gene sequencing[34,35].

Because the sampling of ADs was directed towards mesophilic digesters treating surplus sludge from WWTPs, we also evaluated the MiDAS 5 coverage for ADs treating different primary substrates and temperatures (Supplementary Fig. 2). MiDAS 5 gave very good coverage for all sample types supporting the general applicability of the reference database for ADs. Finally, to provide additional support for the general applicability of the MiDAS 5 database, we evaluated it based on previously published V4-V5 amplicon data from 90 full-scale ADs at 51 municipal WWTPs unrelated to this study[14]. MiDAS 5 contained high-identity hits for 91.8% ± 6.8% of the ASVs, which was higher than for all the other full-length 16S rRNA gene reference databases evaluated (Supplementary Fig. 3).

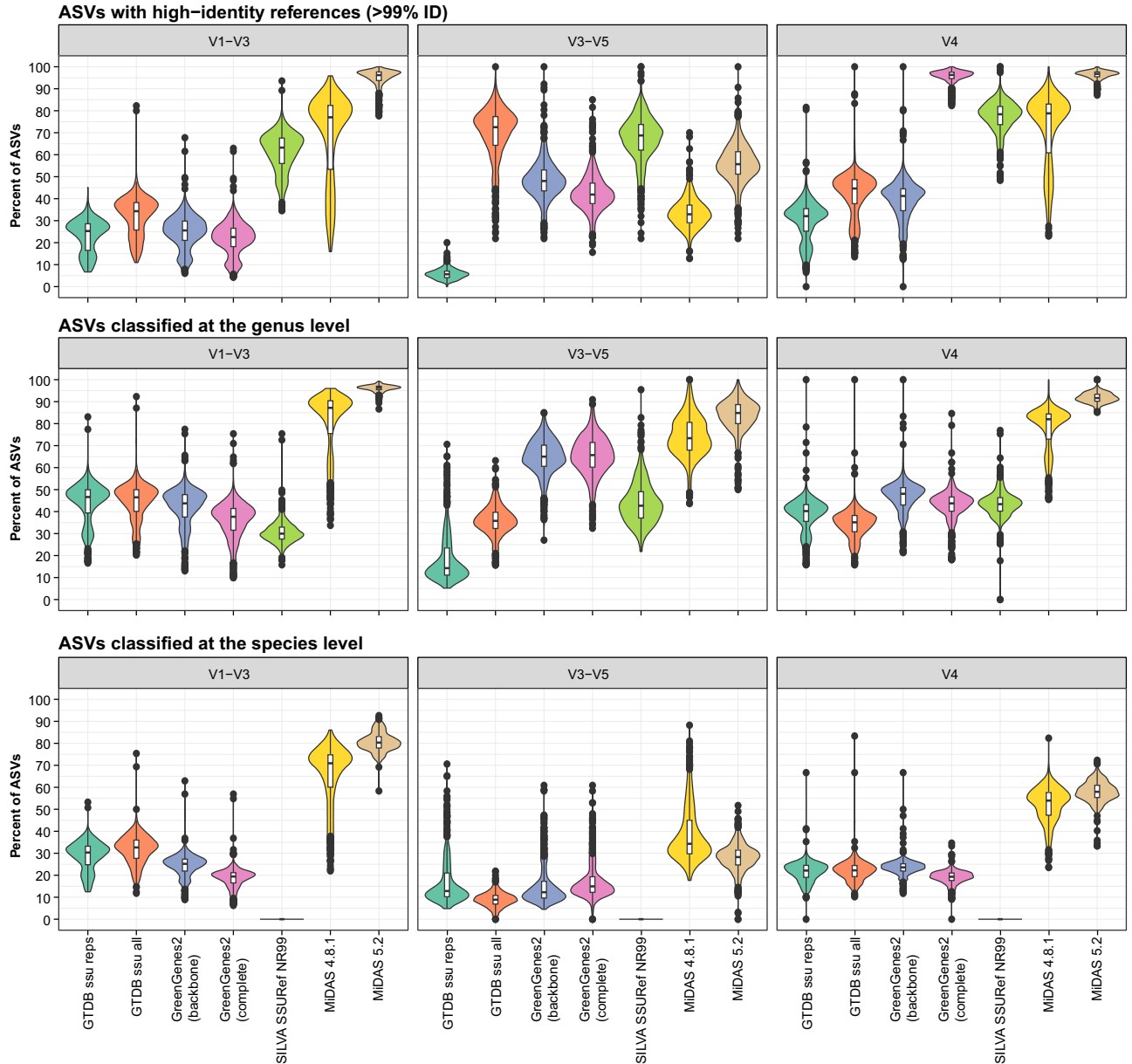

**Fig. 2 | Database evaluation based on short-read amplicon data from this study.** The ASVs for each of the samples were filtered based on their relative abundance (only ASVs with ≥0.01% relative abundance were kept) before the analyses. The percentage of the microbial community represented by the remaining ASVs after the filtering was 95.44% ± 2.23% (mean ± SD) for V1-V3 amplicons (only bacteria), 99.65% ± 0.17% for V3-V5 amplicons (mainly archaea), and 97.34% ± 2.01% for V4 amplicons (bacteria and archaea) across samples. High-identity (≥99%) hits were determined by stringent mapping of ASVs to each reference database. Classification of ASVs was done using the SINTAX classifier. The violin and box plots illustrate the distribution of the percentage of ASVs with high-identity hits or genus/species-level classifications for each database, analyzed across 570 biologically independent samples, including two biological replicates for each digester. Box plots indicate median (middle line), 25th, 75th percentile (box), and the min and max values after removing outliers based on 1.5x interquartile range (whiskers). Outliers have been removed from the box plots to ease visualization. Different colors are used to distinguish the different databases: GTDB_bac120_ssu_reps_r214, GTDB_ssu_all_r214, GreenGenes2_2022_10 (backbone and complete database), SILVA 138.1 SSURef NR99, MiDAS 4.8.1, and MiDAS 5.2.

Our second database evaluation was based on the classification of ASVs from each amplicon dataset using the SINTAX classifier (Fig. 2). We found that MiDAS 5 greatly improved the rates of genus-level classification (96.3% ± 1.4% for V1-V3, 91.5% ± 2.6% for V4, and 82.6% ± 7.5% for V3-V5) compared to MiDAS 4 (80.2% ± 14.9% for V1-V3, 77.3% ± 10.5% for V4, and 74.7% ± 9.3% for V3-V5), and the rates of classification were more than two fold higher than those obtained with any of the other evaluated databases for bacteria and also higher for archaea. Analysis of species-level classifications revealed similar improvements with MiDAS 5 for bacteria (Fig. 2). However, a decrease in species-level classifications was observed between MiDAS 4 and 5 for the archaeal V3-V5 dataset. We hypothesize that this effect relates to over-classifications with MiDAS 4 due to the lack of appropriate reference sequences in this database.

Finally, we investigated if the additional reference sequences introduced in MiDAS 5 could improve classification of amplicon data from WWTPs based on data from the MiDAS global sampling of WWTPs[20] and the Global Water Microbiome Consortium project[36] (Supplementary Fig. 4). Interestingly, no statistically significant improvements were observed. This highlights that most of the added references originated from AD-specific taxa.

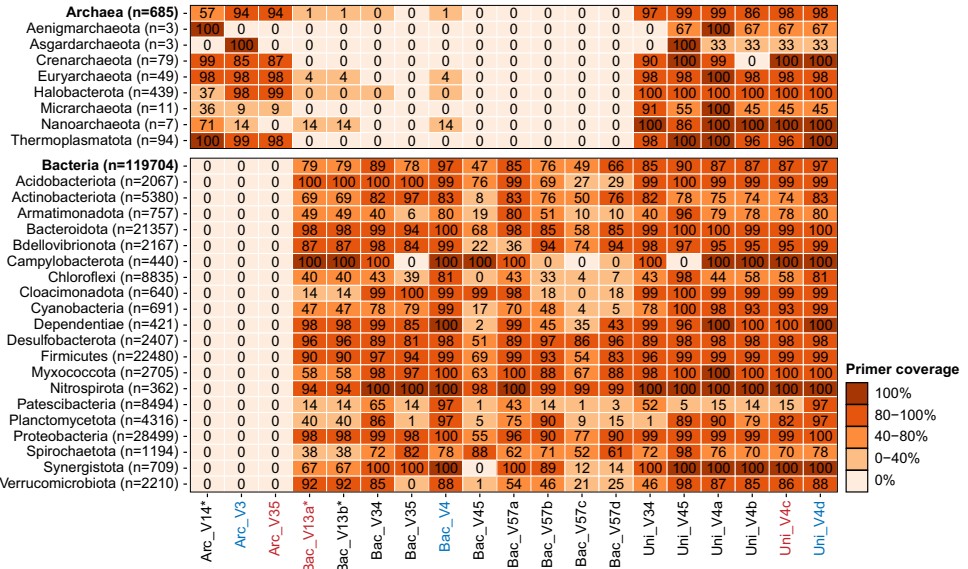

**Fig. 3 | Ecosystem-specific primer coverage for all archaea and bacteria and the 20 most diverse bacterial and all archaeal phyla based on unique FL-ASVs.** The number of FL-ASVs for each taxon ($n$) is provided next to the names. The coverage was determined as the percentage of FL-ASVs in the MiDAS 5.2 database with perfect hits for both forward and reverse primers. The primer pairs marked in red were used in the current study and those marked in blue are recommended by us based on their coverage. Detailed information of all primer pairs and coverage information for all taxa in MiDAS 5.2 are provided in Supplementary Data 2. *Only the reverse primer was evaluated for these primer pairs because the forward primer was used to create the reference sequences in MiDAS. The coverage might therefore be overestimated for these primer pairs.

## Evaluation of 16S rRNA gene amplicon primers for community profiling of ADs

The comprehensive ASV-resolved MiDAS 5 database provides a unique opportunity to determine the theoretical coverage of commonly applied 16S rRNA gene amplicon primer pairs for bacteria and archaea in ADs (Fig. 3). This information is highly valuable when designing experiments, especially if targeting specific taxa. Accordingly, we determined the theoretical coverage for several commonly applied primer pairs for all kingdom to species-level taxa in MiDAS 5 (Supplementary Data 2). We found a fairly low coverage of the V1-V3 primer pair (perfect hits for ≤79% of the bacterial FL-ASVs), which we commonly use due to its high phylogenetic resolution[6,20]. We should therefore expect a significant bias when using this primer pair. The V4 primer used here and in the Earth Microbiome project[37] showed good coverage for both bacteria (perfect hits for 87% of the FL-ASVs) and archaea (perfect hits for 98% of the FL-ASVs). However, a recently published primer pair for the V4 region, designed to improve coverage for Patescibacteria[38], showed even better coverage for bacteria (perfect hits for 97% of the FL-ASVs). Although this primer pair does not target archaea, adding degeneracy at a single base in one of the primers also provided coverage for archaea (perfect hits for 98% of the FL-ASVs). The exceptional coverage offered by this new primer pair leads us to recommend it for the profiling of anaerobic digesters (ADs), despite its lower phylogenetic signal compared to the V1-V3 primers. The V3-V5 primer pair, which was used here to target archaea only, also had good coverage for archaea, though not as good as that of the V4 primers, supporting the choice of the latter.

## Effect of process and environmental factors on the AD microbiota

Alpha diversity analyses showed that the rarefied (10,000 read per sample) ASV richness and inverse Simpsons diversity in ADs were affected mainly by the primary substrate type and the temperature in the ADs (Supplementary Fig. 5). Significantly higher bacterial richness and diversity were observed for ADs treating surplus sludge from WWTPs compared to the other types of substrates. This effect likely reflects the extensive immigration of bacteria into the ADs with the surplus sludge[7,10,39]. A higher richness and diversity were observed for bacteria in mesophilic ADs compared to thermophilic ADs. A similar trend has previously been observed for full-scale ADs treating manure[40,41], household waste[42], and surplus sludge from WWTPs[7].

Genus-level taxonomic beta-diversity was used to investigate the effect of process conditions and geography on the overall microbiota in ADs using principal coordinate analysis (PCoA) and permutational multivariate analysis of variance (PERMANOVA) (Fig. 4). We used this approach because many of the important traits are categorical (yes/no) and only conserved at lower taxonomic ranks (genus/species)[43]. Furthermore, MiDAS 5 enabled us to classify almost all our ASVs at the genus-level, thereby providing a comprehensive description of the microbiota. The PERMANOVA (Adonis $R^2$ values) showed that the overall microbial community was mainly explained by the primary substrate and to a lesser extent by temperature, continent, and digester technology (Fig. 4). This trend was observed for both bacteria and archaea. The percentage of total variation explained by each parameter was, except for the primary substrate, low, suggesting that the global AD microbiota represents a continuous distribution rather than distinct states, as also observed for the human gut microbiota[44] and WWTPs[20]. The pronounced effect of the primary substrates highlights that the overall composition of these substrates are different, but also that some of the feeds contains microbes, particularly in the case of manure and wastewater sludge, which affects the observed diversity in the digesters.

## Core and conditional rare or abundant taxa in the global AD microbiota

The global AD microbiota represents a huge microbial diversity. However, most organisms only occur in very low abundance and are therefore unlikely to have any quantitative impact on the overall metabolism and process performance in ADs. Analysis of core and conditionally rare or abundant taxa (CRAT) is a powerful approach to identify the most important genera and species within a specific ecosystem[20,28,45]. The CRAT may include taxa related to process

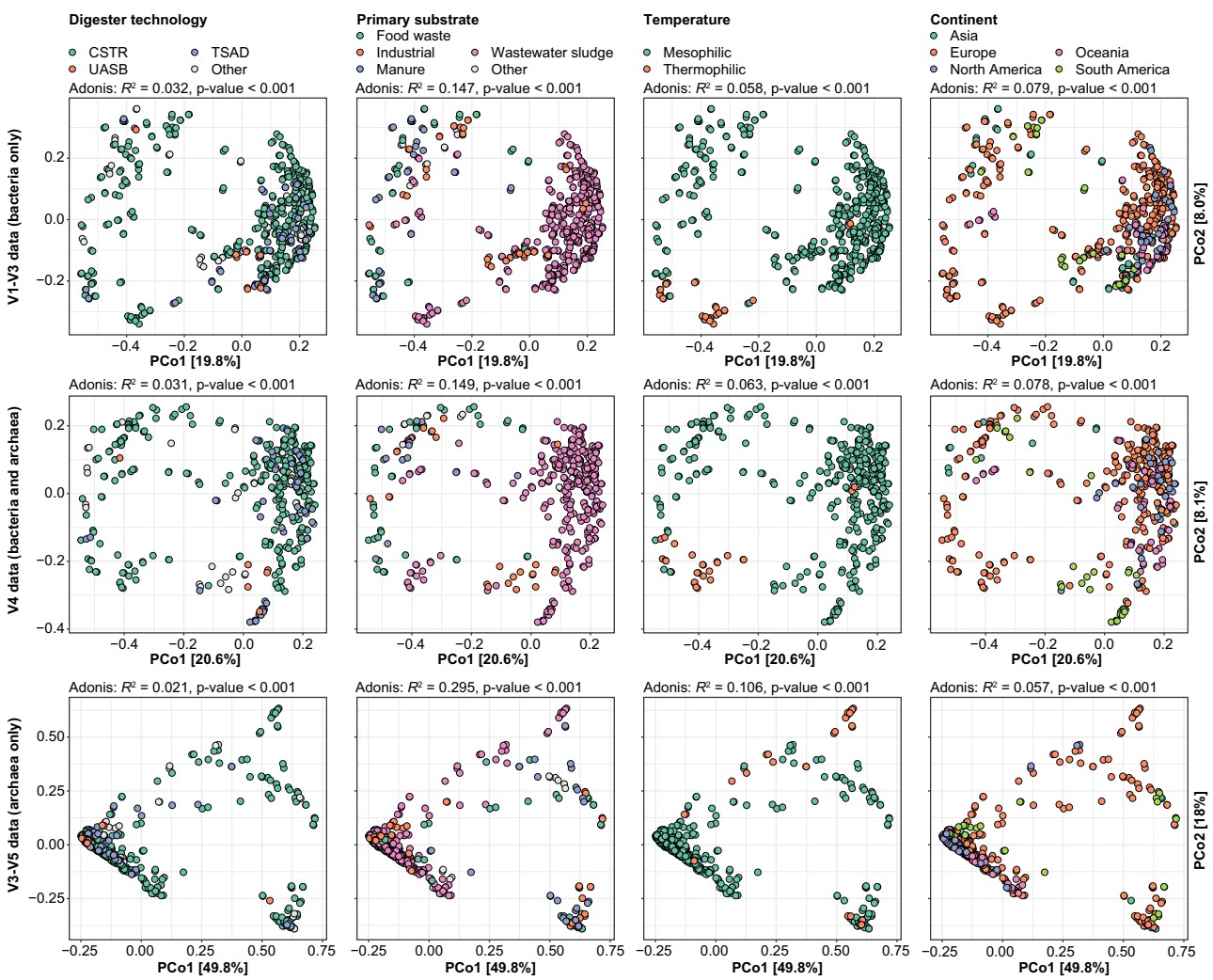

**Fig. 4 | Effects of process and environmental factors on the anaerobic digester microbiota.** Principal coordinate analyses of Bray-Curtis beta-diversity for genera based on V1-V3 (bacteria), V4 (archaea and bacteria), and V3-V5 (archaea) amplicon data. Samples are colored based on metadata. The fraction of variance in the microbial community explained by each variable in isolation was determined by PERMANOVA (Adonis $R^2$-values). Exact $p$-values less than 0.001 could not be confidently determined due to the chosen number of permutations. CSTR Continuous stirred-tank reactor, TSAD Two-stage anaerobic digestion, UASB Upflow anaerobic sludge blanket.

disturbances, such as filamentous microbes associated with foam formation, or taxa associated with the degradation of special substrates found in, e.g., industrial waste.

We recently introduced and applied the following core and CRAT definitions in our survey of the global microbiota of wastewater treatm: strict core (>0.1% relative abundance in >80% of samples), general core (>0.1% relative abundance in >50% of samples), loose core (>0.1% relative abundance in >20% of samples), and CRAT (not part of the core, but present in at least one sample with a relative abundance >1%)[20]. Here, we applied the same criteria to identify core and CRAT genera and species in our global AD dataset. Because the primary substrate showed a strong effect on the overall microbial community (Fig. 4), we determined the core and CRAT for each individual substrate separately (Supplementary Data 3). Only mesophilic ADs were examined for ADs treating food waste, industrial waste, and manure due to the low number of thermophilic ADs sampled. Both mesophilic and thermophilic digesters were examined for ADs treating wastewater sludge. To minimize the impact of primer bias, we analyzed all three amplicon datasets and combined the results, including all core and CRAT that were found in at least one of the datasets.

The core analysis revealed that most core genera were uniquely associated with specific primary substrates and temperature range (Fig. 5a). However, there was also a significant number of core genera shared across substrates (Fig. 5a). In contrast, very few core species were shared between ADs treating different primary substrates (Fig. 5b). This fits well with similar results from a study of ADs in Belgium and Luxemburg[13]. To define a 'most wanted' list for bacteria and archaea in ADs globally, we assigned the highest-ranking category (strict core > general core > loose core > CRAT) across primary substrates, process temperatures, and primer pair to each genus and species (Supplementary Data 3). The resulting list contained 501 core (75 strict, 117 general, and 309 loose) and 191 CRAT genera. The strict core genera included 11 known methanogens and four known syntrophs (*Ca.* Phosphitivorax, *Smithella*, *Syntrophomonas*, *Syntrophorhabdus*). At the species-level, we identified 565 core (29 strict, 126 general, and 410 loose) and 448 CRAT species. The strict core species included two methanogens (*Methanobrevibacter smithii* and *Methanothermobacter* midas_s_3958) and one syntroph (*Syntrophomonas* midas_s_90707). It is worth noting that a large fraction of the taxa observed in ADs does not grow in the digesters, but only occurs because they are in high abundance in the feed[7,10,39]. Previous published data from Danish ADs treating wastewater sludge[7] classified 45 (9.0%) of the core genera observed in this study as non-growing (<20% of ASVs belonging to the specific taxa were classified as

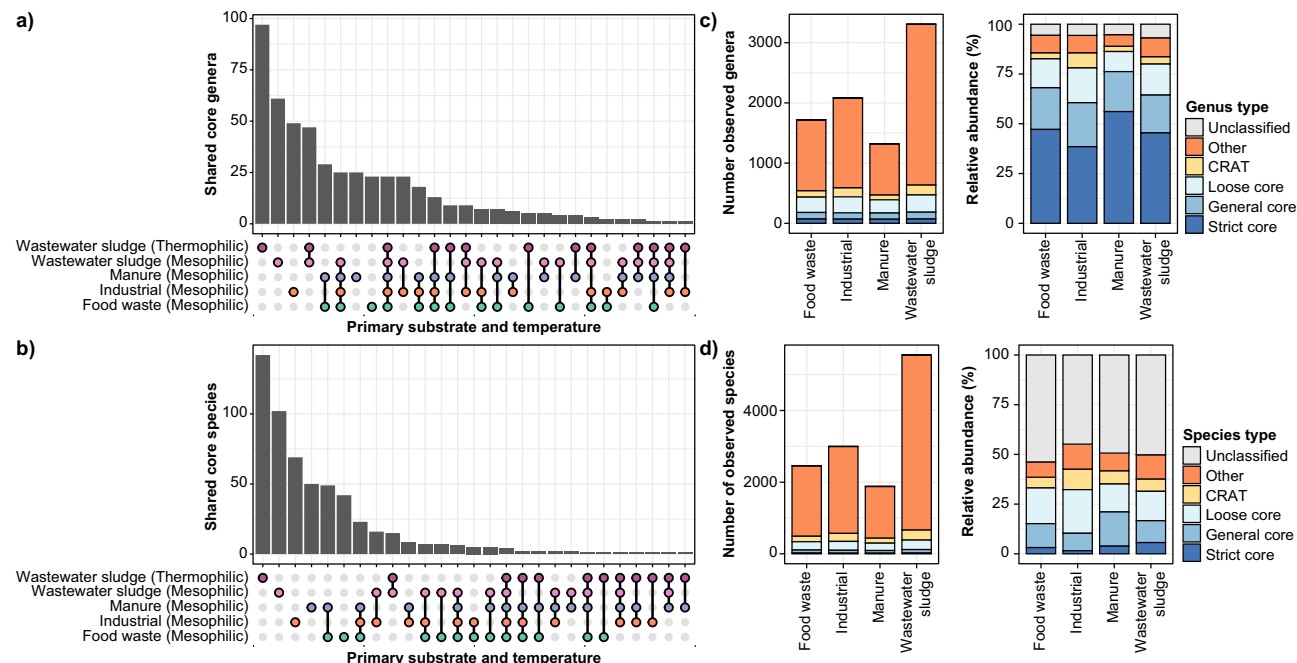

**Fig. 5 | Core and conditionally rare or abundant taxa (CRAT) in anaerobic digesters globally. a, b** UpSet plots displaying the number of shared core genera and species, respectively, across ADs treating different primary substrates and operating at different temperatures. **c, d** Number of observed genera and species, respectively, and their abundance in mesophilic ADs treating different primary substrates based on V4 amplicon data (bacteria and archaea). Values for genera and species are divided into strict core, general core, loose core, CRAT, other taxa, and unclassified ASVs based on the most wanted list (Supplementary Data 3). The relative abundance of different groups was calculated based on the mean relative abundance of individual genera or species across samples. Similar figures for V1-V3 (bacteria only) and V3-V5 (archaea only) amplicons data can be found in Supplementary Fig. 6.

growing), whereas 393 (78.4%) were classified as growing. A similar analysis of core species classified 45 (8.0%) as non-growing and 391 (69.2%) as growing. However, it remains to be determined if these numbers also translate to global ADs.

### Many core and CRAT represent MiDAS placeholder taxa

A large proportion of core and CRAT identified was classified as MiDAS de novo taxa. At the genus-level, 272/501 (54%) of the core genera and 119/191 (62%) of the CRAT genera had only MiDAS placeholder names, and at the species-level, the proportion was even higher. Here placeholder names were assigned to 514/565 (91%) of the core species and 422/448 (94%) CRAT species. These proportions are similar to those observed for the global microbiota in WWTPs[20] and reveals the importance of a taxonomic framework that can handle uncultured taxa which have not yet been officially classified.

### The global AD microbiota is dominated by core and CRAT taxa

Despite only accounting for a minor fraction of the total diversity in the ADs examined, the core and CRAT represented most of the microbes according to relative amplicon read abundance (Fig. 5c, d, Supplementary Fig. 6). The core and CRAT genera accounted for 85-92% (V1-V3), 84-89% (V4), and 96-99% (V3-V5) of the accumulated read abundance in mesophilic ADs depending on primary substrates. The remaining fractions consisted mainly of ASVs unclassified at the genus level, and genera present in very low abundance, presumably with minor importance for the AD performance.

For the species level, the core and CRAT represented 53-61% (V1-V3), 38-43% (V4), and 18-47% (V3-V5) accumulated read abundance depending on the primary substrate. The remaining fractions were mainly composed of ASVs, which could not be classified at the species level, probably due to insufficient phylogenetic resolution of the short-read amplicons[6,31]. The lack of species-level classification was especially pronounced for the archaeal V3-V5 ASVs in ADs treating industrial waste, manure, and wastewater sludge (Supplementary Fig. 6).

The large relative abundance of core and CRAT in the global AD microbiota suggests that we can explain most of the metabolic processes in ADs, if we understand the physiology and metabolic potential of these taxa.

### Global diversity of archaea reveals new potential methanogens

As methanogenic archaea are ultimately responsible for the generation of methane in ADs, we examined the global diversity of archaea in all samples based on the V4 (Fig. 6) and V3-V5 amplicon data (Supplementary Fig. 7). The V4 amplicon data, encompassing both archaea and bacteria, showed that the archaeal reads constituted 5.6% ± 4.4% for ADs treating food waste, 6.8% ± 4.4% for manure, 6.4% ± 2.5% for wastewater sludge, and 13.7% ± 11.1% for industrial waste. Many of the abundant archaea represented well-known methanogens. However, we also observed several abundant genera, only classified based on the MiDAS placeholder taxonomy, affiliated to orders and families of known methanogens. These include midas_g_91627 and midas_g_8154, which represent new families within the orders Methanomicrobiales and Methanofastidiosales, respectively, and midas_g_90473 and midas_g_93310, representing new genera within Methanomassiliicoccaceae and Methanospirillaceae, respectively. In addition, we observed two abundant MiDAS placeholder genera (midas_g_90791 and midas_g_97217) that represent a new order within the class *Ca.* Bathyarchaeia. Members of this class can have a versatile metabolism, and some encode the key methanogenic enzyme methyl-coenzyme M reductase (MCR)[46,47]. Targeted metagenomics and assembly of metagenome-assembled genomes (MAGs) should be applied to confirm the methanogenic potential of these new potential methanogens, and our amplicon datasets provide insight into where these taxa occur in high abundance.

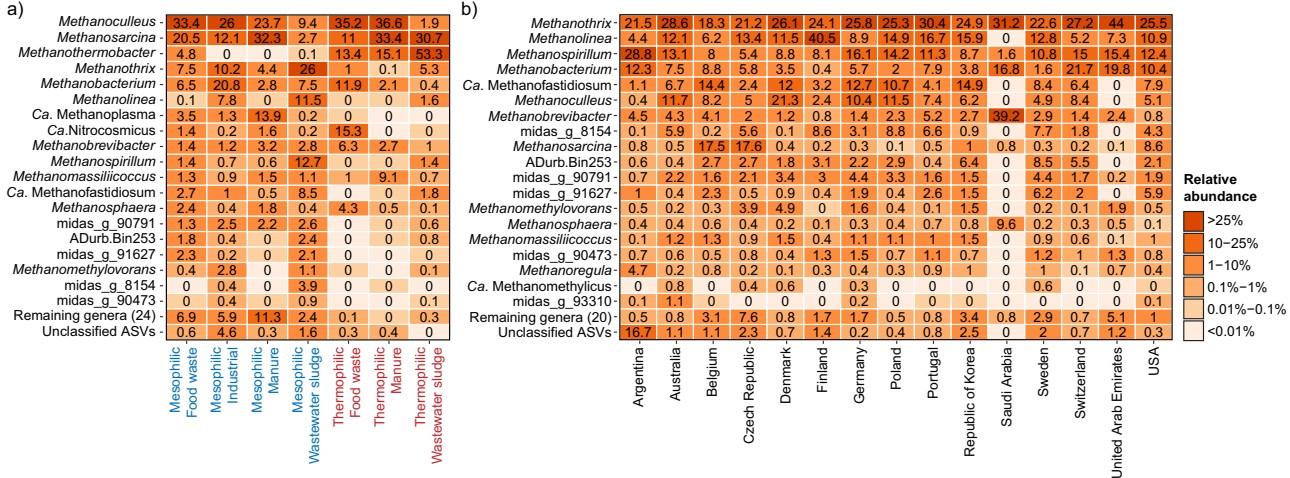

**Fig. 6 | Top 25 archaeal genera based on V4 amplicon data.** The percent relative abundance represents the mean abundance relative to all archaea across (**a**) different temperature range and primary substrates, and (**b**) different countries considering only mesophilic ADs treating mainly wastewater sludge.

The methanogenic community composition was clearly affected by the primary substrate and temperature (Fig. 6a, Supplementary Fig. 7a). The most common methanogens across substrates and temperatures were *Methanoculleus*, *Methanosarcina*, *Methanothermobacter*, and *Methanothrix*. *Methanothermobacter* was as expected most abundant in thermophilic ADs. However, to our surprise, it also occurred in high relative abundance in several mesophilic reactors treating mainly food waste. We were not able to explain their occurrences in these ADs based on the available metadata for the plants, but future studies might shed light on the underlying mechanisms or environmental factors that enable this unexpected distribution.

Because most of our samples originated from mesophilic reactors treating wastewater sludge, we examined the diversity of methanogens across countries in these ADs (Fig. 6b, Supplementary Fig. 7b). This analysis revealed that the same genera were dominating across the world. The most common methanogens in these ADs were *Methanothrix*, *Methanolinea*, *Methanospirillum*, *Methanobacterium*, and the recently discovered *Ca.* Methanofastidiosum[48]. Next, we examined if the methanogens were also conserved at higher phylogenetic resolution. As many archaeal ASVs could not be classified at the species-level, we examined the global diversity at the ASV-level (Supplementary Fig. 8). We found that the vast majority of the abundant ASVs occurred globally. The significant similarity of methanogens across various regions indicates substantial potential for global knowledge transfer concerning their management and utilization.

Among the highly abundant archaea, we also observed an ammonia oxidizing archaeon (AOA) from the genus *Ca.* Nitrosocosmicus[49], which was especially abundant in thermophilic ADs treating food waste. This is surprising and may indicate that they also have an anaerobic physiology which should be investigated further. Another abundant archaeon was the *Ca.* Diapherotrites ADurb.bin253 belonging to the order Woesearchaeales which are characterized by ultra-small genomes and an anaerobic and parasitic/fermentation-based lifestyle[50].

### Global diversity of syntrophic bacteria

Syntrophic bacteria play a vital role in ADs by converting substrates, such as short-chain fatty acids, into acetate, $H_2$, and formate[29,51,52]. These compounds serve as substrates or reducing equivalents for methanogens, which in turn produce methane and $CO_2$. This obligately mutualistic metabolism is crucial because the syntrophs can only oxidize substrates and sustain growth under anaerobic conditions if the methanogens rapidly consume their products to maintain them at

very low concentrations[51,53]. Due to the fastidious metabolism, syntrophs are usually present in low abundance, and can easily become the bottleneck in the anaerobic digestion process[7,8]. Accordingly, we investigated the global diversity of this functional guild in the ADs sampled (Fig. 7, Supplementary Fig. 9).

A clear effect of the primary substrates and digester temperature was observed on the composition and abundance of syntrophic genera in the digesters (Fig. 7a, Supplementary Fig. 9a). The most abundant genus across substrates and temperature was *Syntrophaceticus*, despite being barely detected in ADs treating wastewater sludge. The type strain of this genus, *S. schinkii* Sp3[T], is an acetate-oxidizing syntroph that thrives, and has a competitive advantage, under high ammonium concentrations (up to 8400 mgN/L)[54,55]. The lack of *Syntrophaceticus* in ADs treating wastewater sludge may therefore be explained by lower ammonium concentrations in these ADs (1617 ± 4312 mgN/L, *n* = 145) compared to those treating food waste (2913 ± 1681 mgN/L, *n* = 33), and manure (3449 ± 933 mgN/L, *n* = 18).

*Syntrophomonas*, the second most abundant genus, was common in all AD types investigated, indicating a broader ecological niche. Isolated representatives from this genus can grow syntrophically via β-oxidation of saturated fatty acids of various lengths (C4-C18, depending on the strain)[56–59], and they are therefore likely important for the conversion of long-chain fatty acids in ADs. Among the abundant syntrophs, *Tepidimicrobium*, a member of the order Clostridiales, was also observed in all AD types except mesophilic ADs treating wastewater sludge. The exact metabolism of *Tepidimicrobium* in ADs remains to be determined, however all isolated representatives can degrade proteinaceous compounds and some species can also use carbohydrates[60]. Furthermore, *Tepidimicrobium* has been proposed to grow syntrophically by direct interspecies electron transfer (DIET) with *Methanothermobacter* in a process similar to that observed for *Geobacter*[61]. Accordingly, it is likely that the *Tepidimicrobium* acts as a syntrophic primary degrader in the ADs targeting mainly proteins, carbohydrates, and derivatives.

Finally, we observed a high abundance of the genus *Smithella* in mesophilic ADs treating industrial waste, manure, and wastewater sludge. The type strain *S. propionica* LYP[T] is a propionate oxidizing syntroph, which uses a unique dismutation pathway in which propionate is first converted to acetate and butyrate, and butyrate is hereafter β-oxidized syntrophically to acetate and hydrogen[62,63]. Calculations of Gibbs free energy for this special propionate metabolism indicates a higher tolerance toward elevated hydrogen concentrations[64], which could explain why some *Smithella* prevail in certain ADs. However, *Smithella* has also been implicated in the syntrophic degradation of

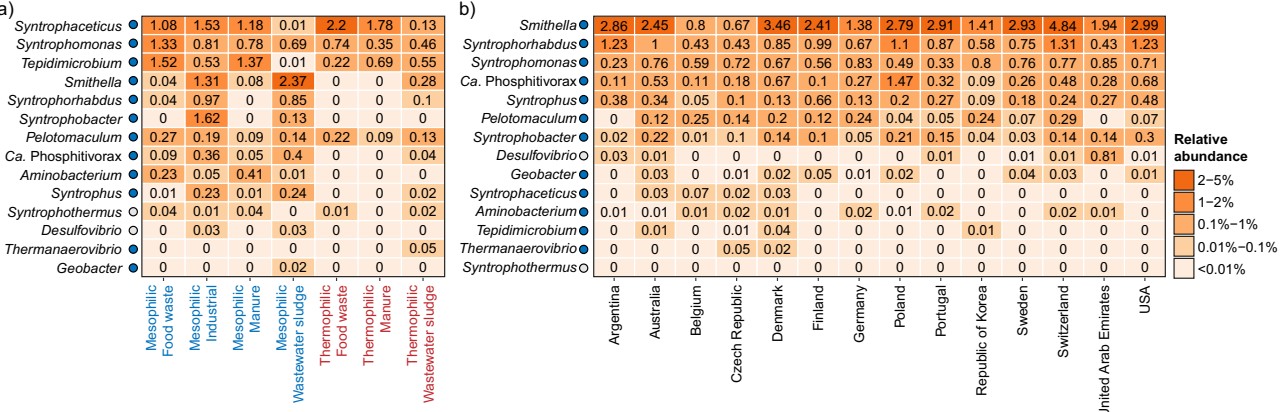

**Fig. 7 | Global diversity of syntrophs based on V1-V3 amplicon data.** The percent relative abundance represents the mean for genera across (**a**) different temperature range and primary substrates, and (**b**) different countries considering only mesophilic ADs treating mainly wastewater sludge. Colored circles next to the genus labels indicate whether the genera have previously been identified as growing in ADs at Danish WWTPs according to Jiang et al.[7]. Blue: >50% of ASVs classified as growing; Yellow: 20–50% of ASVs classified as growing. Red: <20% of ASVs classified as growing. Gray: No information available for the specific genus.

long-chain alkanes[65,66], which could reflect a more versatile metabolism.

When investigating geographical diversity of syntrophic fatty acid oxidizing bacteria in mesophilic ADs treating wastewater sludge, a similar pattern was observed across countries (Fig. 7b, Supplementary Fig. 9b). *Smithella*, was generally the dominating syntroph. However, *Syntrophomonas*, *Syntrophorhabdus*, *Ca.* Phosphitivorax, and *Syntrophus* also occurred at a high relative abundance in almost all countries. Isolates of *Syntrophorhabdus*, including the type strain *S. aromaticus* UI[T], are syntrophic fermenters of aromatic compounds and may accordingly play an important role in the detoxification of these substrates in ADs[67,68]. *Ca.* Phosphitivorax was recently discovered as a butyrate degrading syntroph by genome-resolved meta-transcriptomics in a digester treating wastewater sludge[52], and *Syntrophus* participates in the degradation of fatty acids and aromatics[69,70]. Overall, the results suggest a complex syntrophic degradation process, which involves multiple genera with different substrate specificities.

To gain additional insight into the global diversity of syntrophs, we also investigated the species-level diversity across mesophilic digesters treating wastewater sludge (Supplementary Fig. 10). We observed a large species diversity among most of the abundant syntrophic genera. Furthermore, we found that the most abundant species in the ADs were often distinct from the isolated representatives, which prompts for further investigations into the metabolic potential of syntrophs in situ.

### Global diversity of filamentous bacteria

Foaming is a common operational problem in ADs and has a strong negative impact on process performance resulting in considerable costs. Both abiotic and biotic factors are involved in foaming[71]. The abiotic factors include high loading rates of surfactants (oil, grease, fatty acids, detergent, proteins, and particulate matter) and biosurfactants produced by microbes in the digester[72]. The biotic factors cover increased abundance of hydrophobic, filamentous microorganisms that can interact with, and stabilize, gas bubbles in the foam[71,73]. To gain further insight into potential foam forming microbes, we examined the global diversity of known filamentous bacteria in ADs (Fig. 8, Supplementary Fig. 11).

The diversity and mean relative abundance of known filamentous organisms were generally low in the ADs examined except for those treating wastewater sludge (Fig. 8a, Supplementary Fig. 11). However, the increased diversity and abundance in the latter are to a large extent the result of passive immigration from the fed surplus sludge.

However, most of these are likely unable to grow in the ADs[7]. *Anaerolinea*, *Ca.* Brevefilum, and *Trichococcus* were common across ADs treating all primary substrates (Fig. 8a, Supplementary Fig. 11), whereas *Ca.* Microthrix and *Ca.* Promineofilum were mainly observed in ADs treating wastewater sludge. Many of the Chloroflexi genera found here were also observed in a recent meta-analysis of amplicon data from 17 studies representing 62 ADs[74]. Several of the abundant filamentous genera, including *Ca.* Microthrix and *Ca.* Brevefilum, were previously found to correlate with the foaming potential of full-scale digester sludge from mesophilic ADs at WWTPs[73]. *Ca.* Brevefilum seems especially interesting as it grows well in ADs[7,75].

The species-level diversity was generally low for the filamentous bacteria (Supplementary Fig. 12). *Ca.* Brevefilum was dominated by *Ca.* B. fermentans, *Trichococcus* by midas_s_4, *Ca.* Microthrix by *Ca.* M. parvicella and *Ca.* M. subdominans, and *Gordonia* by *G. defluvii* and *G. amarae*. *Ca.* Promineofilum was dominated by *Ca.* P. glycogenico, but a few MiDAS placeholder species, were also commonly observed. The low species-level diversity of potential foam-forming bacteria suggests that it may be feasible to develop and implement universal mitigation strategies for these bacteria in ADs worldwide.

### Final remarks and perspectives

MiDAS 5 was made possible thanks to a huge collaborative effort from experts worldwide, who contributed to the project by sampling and providing metadata for ADs in their respective countries. Building on the success of its predecessor, MiDAS 4, this latest expansion covers ASV-resolved, full-length 16S rRNA gene references from numerous ADs from all parts of the globe covering different operation parameters and different substrates. This expanded database provides greatly improved coverage for AD-specific taxa and a strongly needed taxonomy for uncultured lineages, which lack official taxonomic classification. As such, it will be an invaluable resource for researchers and AD professionals, providing them with a common point of reference to facilitate knowledge sharing and pave the way for a comprehensive understanding of the AD microbiome.

Our in silico 16S rRNA gene primer evaluation based on the MiDAS 5 database revealed that the coverage of commonly applied primer pairs varies significantly, with some having low coverage and potential bias towards certain taxa. Because the primer coverage was evaluated for all taxa in the MiDAS 5 database and at all taxonomic ranks, it provides a solid foundation for designing experiments and targeting specific taxa in future studies. For general microbial profiling of ADs, we would recommend the use of the newly improved universal V4 primer pair[38], as it show excellent

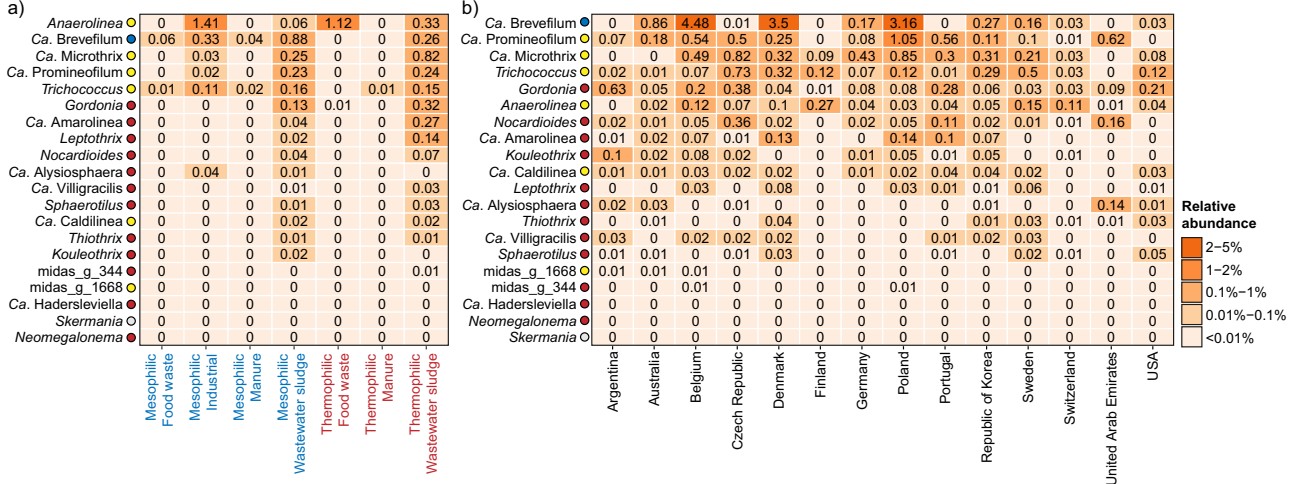

**Fig. 8 | Global diversity of known filamentous bacteria based on V1-V3 amplicon data.** The percent relative abundance represents the mean for genera across (**a**) different temperature range and primary substrates, and (**b**) different countries considering only mesophilic ADs treating mainly wastewater sludge. Colored circles next to the genus labels indicate whether the genera have previously been identified as growing in ADs at Danish WWTPs according to Jiang et al.[7]. Blue: >50% of ASVs classified as growing; Yellow: 20–50% of ASVs classified as growing; Red: <20% of ASVs classified as growing. Gray: No information available for the specific genus.

coverage for both archaea and bacteria in both WWTPs and the AD ecosystem.

Although the total microbial diversity in ADs is huge, importantly, we showed that less than 1000 genera and species accounted for most of the microbes in the AD ecosystem. By focusing on the fraction of these abundant and common microbes that can grow in the AD systems, we will be able to explain most of the microbial processes that occur in the anaerobic digestion process. This list of "Most Wanted" organisms contain species that should be prime targets for future in situ studies and the reconstruction of MAGs. These genomes can then be annotated to provide additional details about their potential metabolic pathways and roles in the AD ecosystem[15,16,76–78].

The global survey of the AD microbiota using three different primer pairs provided a unique insight into the global diversity of individual AD taxa and clues into the environmental and operational factors that define their ecological niches. This information will be invaluable in the development of future microbiome management strategies and improved sustainability of the field of anaerobic digestion.

To enhance knowledge dissemination, we have updated the MiDAS Field Guide available at www.midasfieldguide.org. This dynamic resource allows users to delve into specifics related to the physiology, morphology, and ecology of genera listed in the MiDAS database. Additionally, it offers country-specific data on the prevalence of all MiDAS genera and species in WWTPs and ADs. Finally, it provides information on the availability of fluorescence in situ hybridization probes and reference genomes, paving the way for subsequent research endeavors.

## Methods
### Sampling and metadata collection
To facilitate sampling of ADs worldwide, we established the MiDAS Global Consortium for Anaerobic Digesters, which consists of 25 anaerobic digestion experts in 19 countries. Members of the consortium acted as national sampling coordinators and were in direct contact with the ADs. Two samples were obtained from each AD and shipped on ice to the sampling coordinators. For each replicate, 2 mL sample was preserved in 2 mL RNAlater (Invitrogen), stored at 4 °C until all national samples were collected (usually within a few days), and then shipped to Aalborg University with cooling elements. Upon

arrival, the samples were separated into aliquots that were prepared for nucleic acid purification. Metadata associated with each AD was also obtained by the sampling coordinators and is provided as Supplementary Data 1. Minimum information from all ADs included continent, country, GPS coordinates, sampling date, temperature in the digester ("Mesophilic" (≤45 °C) or "Thermophilic" (50-60 °C)), primary substrate ("Wastewater sludge", "Industrial", "Food waste", "Manure", or "Other"), and digester technology ("Two-stage digester (TSAD)", "Continuous Stirred-tank Reactor (CSTR)", "Upflow anaerobic sludge blanket (UASB)", or "Other").

### General molecular methods
All commercial kits were used according to the protocols provided by the manufacturer unless otherwise stated. The concentration and quality of nucleic acids were determined using a Qubit 3.0 fluorometer (Thermo Fisher Scientific) and an Agilent 2200 Tapestation (Agilent Technologies), respectively.

### Nucleic acid purification
DNA was purified using a custom plate-based extraction protocol based on the FastDNA spin kit for soil (MP Biomedicals). The protocol is available at www.midasfieldguide.org (aau_ad_dna_v 2.0). RNAlater preserved samples were thawed and homogenized using a Heidolph RZR 2020 laboratory stirrer. 20 μL of sample was resuspended in 300 μL PBS and transferred to Lysing Matrix E barcoded tubes (MP Biomedicals). 40 μL of MT buffer was added and lysis was performed by bead beating in a FastPrep-96 bead beater (MP Biomedicals) (3 × 120 s, 1800 rpm with 2 min incubation on ice between cycles). The samples were centrifuged (3486 ×g, 10 min) and 200 μL supernatant was transferred to a 96-well PCR-plate. 50 μL Protein Precipitation Solution (PPS) was mixed with each sample, which was then centrifuged again. 150 μL supernatant was cleaned-up using 100 μL CleanNGS beads with elution into 60 μL of nuclease-free water. 40 μL of the purified DNA was transferred to a new 96-well plate and stored at -80 °C.

### Full-length 16S rRNA gene library preparation, sequencing, and processing
Full-length 16S rRNA gene sequencing was carried out using high-accuracy, long-read amplicon sequencing using unique molecular

identifiers (UMIs) and PacBio circular consensus sequencing (CCS)[79]. Oligonucleotides used can be found in Supplementary Table 1. Bacterial and archaeal 16S rRNA genes were UMI-tagged using overhang primers based on the 27F and 1391R[80] and SSU1ArF and SSU1000ArR[34] primer pairs, respectively. These primers have shown excellent coverage for the known bacterial and archaeal diversity in silico[34,80].

**Addition of UMI-tags by overhang PCR.** Adaptors containing UMIs, and defined primer binding sites were added to each end of the bacterial and archaeal 16S rRNA genes by PCR. The reaction contained 20 μL of 5x SuperFi Buffer (Invitrogen), 2 μL of 10 mM dNTP mix, 5 μL of 10 μM f16S_pcr1_fw, 5 μL of 10 μM f16S_pcr1_rv, 1 μL of 2 U/μL Platinum SuperFi DNA polymerase (Invitrogen), 100 ng of pooled template DNA (from all ADs), and nuclease-free water to 100 μL. The reaction was incubated with an initial denaturation at 98 °C for 30 s followed by 2 cycles of denaturation at 98 °C for 20 s, annealing at 55 °C for 30 s, and extension at 72 °C for 45 s, and then a final extension at 72 °C for 5 min. The sample was purified using 0.6x CleanNGS beads and eluted in 20 μL nuclease-free water.

**Primary library amplification.** The tagged 16S rRNA gene amplicons were amplified using PCR to obtain enough product for quantification. The reaction contained 19 μL of UMI-tagged sample, 20 μL 5x SuperFi buffer (Invitrogen), 2 μL of 10 mM dNTP, 5 μL of 10 μM f16S_pcr2_fw, 5 μL of 10 μM f16S_pcr2_rv, 48 μL nuclease-free water, and 1 μL 2U/μL Platinum SuperFi DNA polymerase (Invitrogen). The reaction was incubated with an initial denaturation at 98 °C for 30 s followed by 15 cycles of denaturation at 98 °C for 20 s, annealing at 60 °C for 30 s, and extension at 72 °C for 45 s and then a final extension at 72 °C for 5 min. The PCR product was purified using 0.6x CleanNGS beads and eluted in 11 μL nuclease-free water. The amplicons were validated on a Genomic screentape and quantified with the Qubit dsDNA HS assay kit.

**Clonal library amplification.** Tagged amplicon libraries were diluted to ~250,000 molecules/μL and amplified by PCR to obtain clonal copies of each uniquely tagged amplicon molecule. Three libraries were made for the bacterial 16S rRNA genes and one for archaea. The PCR reactions contained 1 μL diluted primary library, 20 μL 5x SuperFi buffer (Invitrogen), 2 μL of 10 mM dNTP, 5 μL of 10 μM f16S_pcr2_fw, 5 μL of 10 μM f16S_pcr2_rv, 66 μL nuclease-free water, and 1 μL 2U/μL Platinum SuperFi DNA polymerase (Invitrogen). The reaction was incubated with an initial denaturation at 98 °C for 30 s followed by 25 cycles of denaturation at 98 °C for 20 s, annealing at 60 °C for 30 s, and extension at 72 °C for 45 s and then a final extension at 72 °C for 5 min. The PCR product was purified using 0.6x CleanNGS beads and eluted in 20 μL nuclease-free water. The amplicons were validated on a Genomic screentape and quantified with the Qubit dsDNA HS assay kit.

**PacBio CCS sequencing.** The four clonal libraries were sent to Admera Health (Plainfield, NJ, USA) for PacBio library preparation and sequencing. Here amplicons were incubated with T4 polynucleotide kinase (New England Biolabs) following the manufacturer's instructions, and sequencing library prepared using SMRTbell Express Template Preparation kit 1.0 following the standard protocol. Sequencing was performed using 4x SMRT cells on a Sequel II using a Sequel II Sequencing kit 1.0, Sequel II Binding and Int Ctrl kit 1.0 and Sequel II SMRT Cell 8 M, following the standard protocol with 1 h pre-extension and 15 h collection time (Pacific Biosciences).

**Bioinformatic processing.** CCS reads were generated from raw PacBio data using CCS v.3.4.1 (https://github.com/PacificBiosciences/ccs) with default settings. UMI consensus sequences (consensus_raconx3.fa) were obtained using the longread_umi script (https://github.com/SorenKarst/longread_umi)[79] using the following options: pacbio_pipeline, -v 3, -m 1000, -M 2000, -s 60, -e 60, -f CAAGCAGAAGACGG

CATACGAGAT, -F AGRGTTYGATYMTGGCTCAG (bacteria) or TCCG GTTGATCCYGCBRG (archaea), -r AATGATACGGCGACCACCGAGATC, -R GACGGGCGGTGWGTRCA (bacteria) or GGCCATGCAMYWCCTCTC (archaea), and -c 3. The UMI-consensus reads were oriented based on the SILVA 138.1 SSURef NR99 database using the usearch v.11.0.667 -orient command and trimmed between the 27F and 1391R (bacteria) or SSU1ArF and SSU1000ArR (archaea) primer binding sites using the trimming function in CLC genomics workbench v. 20.0. Sequences without both primer binding sites were discarded. The trimmed high-fidelity reads were processed with AutoTax v. 1.7.4[6] to create FL-ASVs and these were added to the MiDAS 4.8.1 reference database[20] to create MiDAS 5.0. Subsequent updates to MiDAS 5.2 were made to accommodate taxonomic updates (see the release change logs for details).

### Short-read amplicon sequencing

V1-V3 amplicons were made using the 27F (5′-AGAGTTT-GATCCTGGCTCAG-3′)[81] and 534R (5′-ATTACCGCGGCTGCTGG-3′)[82] primers with barcodes and Illumina adaptors (IDT)[83]. 25 μL PCR reactions in duplicate were run for each sample using 1X PCRBIO Ultra Mix (PCR Biosystems), 400 nM of both forward and reverse primer, and 10 ng template DNA. PCR conditions were 95 °C, for 2 min followed by 20 cycles of 95 °C for 20 s, 56 °C for 30 s, and 72 °C for 60 s, followed by a final elongation at 72 °C for 5 min. PCR products were purified using 0.8x CleanNGS beads and eluted in 25 μL nuclease-free water.

V3-V5 amplicons were made using the Arch-340F (5′-CCCTAHGGGGYGCASCA-3′) and Arch-915R (5′-GWGCYCCCCCGY-CAATTC-3′) primers[84]. 25 μL PCR reactions in duplicate were run for each sample using 1X PCRBIO Ultra Mix (PCR Biosystems), 400 nM of both forward and reverse primer, and 10 ng template DNA. PCR conditions were 95 °C, for 2 min followed by 30 cycles of 95 °C for 15 s, 55 °C for 15 s, and 72 °C for 50 s, followed by a final elongation at 72 °C for 5 min. PCR products were purified using 0.8x CleanNGS beads and eluted in 25 μL nuclease-free water. 2 μL of purified PCR product from above was used as template for a 25 μL Illumina barcoding PCR reaction containing 1x PCRBIO Reaction buffer, 1 U PCRBIO HiFi Polymerase (PCR Biosystems) and 10 μL of Nextera adaptor mix (Illumina). PCR conditions were 95 °C, for 2 min, 8 cycles of 95 °C for 20 s, 55 °C for 30 s, and 72 °C for 60 s, followed by a final elongation at 72 °C for 5 min. PCR products were purified using 0.8x CleanNGS beads and eluted in 25 μL nuclease-free water.

V4 amplicons were made using the 515F (5′-GTGYCAGCM GCCGCGGTAA-3′)[82] and 806R (5′-GGACTACNVGGGTWTCTAAT-3′)[85] primers. 25 μL PCR reactions in duplicate were run for each sample using 1X PCRBIO Ultra Mix (PCR Biosystems), 400 nM of both forward and reverse primer, and 10 ng template DNA. PCR conditions were 95 °C, for 2 min followed by 30 cycles of 95 °C for 15 s, 55 °C for 15 s, and 72 °C for 50 s, followed by a final elongation at 72 °C for 5 min. PCR products were purified using 0.8x CleanNGS beads and eluted in 25 μL nuclease-free water. 2 μL of purified PCR product from above was used as template for a 25 μL Illumina barcoding PCR reaction as described for the V3-V5 amplicons.

16S rRNA gene V1-V3, V3-V5, and V4 amplicon libraries were pooled separately in equimolar concentrations and diluted to 4 nM. The amplicon libraries were paired-end sequenced (2 × 300 bp) on the Illumina MiSeq using v3 chemistry (Illumina, USA). 10-20% PhiX control library was added to mitigate low diversity library effects.

### Processing of short-read amplicon data

Usearch v.11.0.667[86] was used for processing of 16S rRNA gene amplicon data and for read mapping. V1-V3 forward and reverse reads were merged using the usearch -fastq_mergepairs command, filtered to remove phiX sequences using usearch -filter_phix, and quality filtered using usearch -fastq_filter with -fastq_maxee 1.0. Dereplication was performed using -fastx_uniques with -sizeout, and amplicon

sequence variants (ASVs) were resolved using the usearch -unoise3 command[87]. An ASV-table was created by mapping the quality filtered reads to the ASVs using the usearch -otutab command with the -zotus and -strand plus options. Taxonomy was assigned to ASVs using the usearch -sintax command with -strand both and -sintax_cutoff 0.8 options. Mapping of ASVs to reference databases was done with the usearch -usearch_global command and the -id 0, -maxaccepts 0, -maxrejects 0, -top_hit_only, and -strand plus options.

16S rRNA gene V3-V5 forward reads (reverse reads in relation the 16S rRNA gene) were filtered to remove phiX sequences using usearch -filter_phix, trimmed to remove primers and obtain a fixed length of 250 bp using -fastx_truncate with -stripleft -17 and trunclen 250, reverse complemented with usearch -fastx_revcomp, and quality filtered using usearch -fastq_filter with -fastq_maxee 1.0. Subsequent processing was like that for the V1-V3 amplicons.

16S rRNA gene V4 forward reads (reverse reads in relation the 16S rRNA gene) were trimmed with cutadapt v.2.8[88] based on the V4 primers with the -g ^GGACTACHVGGGTWTCTAAT...TTACCGCGGCK GCTGGCAC and --discard-untrimmed options. The trimmed reads, which span the entire V4 amplicon, were reverse complemented with usearch -fastx_revcomp, and quality filtered using usearch -fastq_filter with -fastq_maxee 1.0. Subsequent processing was like that for the V1-V3 amplicons.

### In silico primer evaluation

The specificity of commonly used amplicon primers was determined for each FL-ASV using the analyze_primers.py script from Primer Prospector v. 1.0.1[89]. The specificity of primer sets was defined based on the overall weighted scores (OWS) for the primer with the highest score as follows: Perfect hit (OWS = 0), partial hit (OWS > 0, and ≤1), poor hit (OWS > 1). The percentage of perfect hits were calculated in R for all taxa in MiDAS 5.

### Microbial community analyses

Short-read amplicon data was analyzed with R v.4.3.2[90] through RStudio IDE v.2023.12.1[91], with the tidyverse v.2.0.0 (https://www.tidyverse.org/), vegan v.2.6-4[92], maps v.3.4.2[93], data.table v.1.14.10[94], FSA v.0.9.5[95], rcompanion v. 2.4.35[96], patchwork v.1.1.3[97], ggupset v.0.3.0[98] and Ampvis2 v.2.8.6[99] packages.

The microbial community analyses were performed based on all three 16S rRNA gene short-read amplicon dataset (V1-V3, V3-V5, and V4). Samples with <10,000 reads and those lacking information about digester technology, primary substrate, and temperature in the digester were discarded from the analyses. After filtration, 547 V1-V3, 542 V3-V5, and 430 V4 samples remained.

Associations between the AD microbiota and the following process-related or environmental variables were investigated: Digester technology, primary substrate, temperature in the digester, and continent (see definitions above). All variables were treated as factors.

For alpha diversity analyses, samples were rarefied to 10,000 reads, and alpha diversity (observed ASVs and inverse Simpson) was calculated using the ampvis2 package. The Kruskal-Wallis with Dunn's post-hoc test (Bonferroni correction with α = 0.01 before correction) was used to determine statistically significant differences in alpha diversity between samples grouped by process and environmental variables.

Beta diversity distances based on Bray-Curtis (abundance-based) for genera was calculated using the vegdist function in the vegan R package and visualized by PCoA plots with the ampvis2 package. To determine how much individual parameters affected the structure of the microbial community across the ADs, a permutational multivariate analysis of variance (PERMANOVA) test was performed on the beta-diversity matrices using the adonis function in the vegan package with 999 permutations.

Core taxa (genera and species) were determined separately for ADs treating different primary substrates and operating at different temperatures (mesophilic and thermophilic) based on their relative abundances in individual ADs according to the three short-read amplicon datasets. Core taxa definitions were identical to those applied in the MiDAS global survey of WWTPs[20]. Taxa were considered abundant when present at >0.1% relative read abundance in individual ADs. Based on how frequently taxa were observed to be abundant, we defined the following core communities: loose core (>20% of ADs), general core (>50% of ADs), and strict core (>80% of ADs). Additionally, we defined conditionally rare or abundant taxa (CRAT)[100] composed of taxa present in one or more ADs at >1% relative abundance, but not belonging to the core taxa.

### Reporting summary

Further information on research design is available in the Nature Portfolio Reporting Summary linked to this article.

## Data availability

The raw and assembled sequencing data generated in this study have been deposited in the NCBI SRA database under accession code PRJNA1019951. The MiDAS 5 reference database in SINTAX, QIIME and DADA2 format is available at the MiDAS fieldguide website [https://www.midasfieldguide.org/guide/downloads].

## Code availability

R scripts used for data analyses and figures are available at GitHub [https://github.com/msdueholm/MiDAS5][101]. Raw data files for the R scripts are available at Figshare [https://doi.org/10.6084/m9.figshare.24219199.v2][102].

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

## Acknowledgements

The project has been funded by the Danish Research Council (grant 6111-00617 A, P.H.N.) and the Villum Foundation (Dark Matter and grant 13351, P.H.N.). We thank all the involved anaerobic digester plants for providing samples and plant metadata.

## Author contributions

P.H.N. and M.K.D.D. designed the study. M.K.D.D. and P.H.N. wrote the manuscript and all authors reviewed and approved the final manuscript. M.A., Y.B-F., D.B., C.B., M.C.C, Å.D., L.E., C.H., K.K., N.K., C.L., G.L., S.M., V.O., P.O-P., D.P., V.R., M.R., J. Rajal., P.E.S., N.T., J.V., J.D.V., C.W. provided samples and metadata. V. Rudkjøbing. handled sampling, DNA extraction and library preparation for DNA sequencing. M.K.D.D. and K.S.A. performed the bioinformatics analyses. M.K.D.D., A-K.C.K. and K.S.A. curated metadata and carried out statistical analyses.

## Competing interests

The authors declare no competing interests.

## Additional information

¹Center for Microbial Communities, Department of Chemistry and Bioscience, Aalborg University, Aalborg, Denmark. ²Centre of Biological Engineering, University of Minho, Minho, Portugal. ³School of Water, Energy and Environment, Cranfield University, Cranfield, UK. ⁴Australian Centre for Water and Environmental Biotechnology (ACWEB), The University of Queensland, Brisbane, Australia. ⁵Department of Civil and Environmental Engineering, University of Massachusetts Amherst, Amherst, MA, USA. ⁶Instituto de Investigaciones para la Industria Química (INIQUI), Universidad Nacional de Salta (UNSa) - Consejo Nacional de Investigaciones Científicas y Técnicas (CONICET), Salta, Argentina. ⁷Department of Chemical Engineering, Lund University, Lund, Sweden. ⁸INGEBI-CONICET, University of Buenos Aires, Buenos Aires, Argentina. ⁹Laboratory for Environmental Biotechnology, Ecole Polytechnique Fédérale de Lausanne (EPFL), Lausanne, Switzerland. ¹⁰Chair of Urban Water Systems Engineering, Technical University of Munich (TUM), Garching, Germany. ¹¹Institute of Water Quality and Resource Management, TU Wien, Vienna, Austria. ¹²Department of Civil, Urban, Earth, and Environmental Engineering & Graduate School of Carbon Neutrality, Ulsan National Institute of Science and Technology (UNIST), Ulsan, South Korea. ¹³School of Chemical Engineering, National Technical

University of Athens, Zografou, Greece. [14]Applied Environmental Biotechnology Laboratory, Birla Institute of Technology and Science (BITS-Pilani), Pilani, Goa campus, Goa, India. [15]School of Biological and Chemical Sciences and Ryan Institute, University of Galway, Galway, Ireland. [16]Water Supply and Bioeconomy Division, Faculty of Environmental Engineering and Energy, Poznan University of Technology, Poznan, Poland. [17]Department of Water Technology and Environmental Engineering, University of Chemistry and Technology Prague, Prague, Czech Republic. [18]Research Scientist at Kemira Oyj, Espoo R&D Center, Espoo, Finland. [19]Chemical Engineering Department, Khalifa University, Khalifa, UAE. [20]Environmental Science and Engineering Program, Biological and Environmental Science and Engineering Division, King Abdullah University of Science and Technology (KAUST), Thuwal, Kingdom of Saudi Arabia. [21]Center for Microbial Ecology and Technology (CMET), Ghent University, Ghent, Belgium. ✉e-mail: md@bio.aau.dk; phn@bio.aau.dk

