## [Peer Review File · Nature Communications]

MiDAS 5: Global diversity of bacteria and archaea in anaerobic digestersReviewers' Comments:

Reviewer #1:

Remarks to the Author:

This paper reported on the establishment of the MiDAS 5 database, which was based on the sequencing of over 500000 high-quality, full-length 16S rRNA gene sequences from 285 full-scale ADs around the world, obtaining a large amount of data, thereby expanding our understanding of the diversity and function of bacteria and archaea in AD. The author used three sets of common primers targeting different regions of the 16S rRNA gene in bacteria and/or archaea to conduct amplicon based global scale microbial community analysis of sampled AD. It also reveals how environmental conditions and biogeography shape the AD microbiota.

This work has made some exciting new breakthroughs on the basis of their original database, but there are still some minor points that need to be addressed before publication.

Specific Comments

12.Line 315-316 I think there should be a reference after this statement.

Line 337-347: This section mainly describes the definitions of certain specific nouns, and it is recommended to reduce or move them up to the introduction section.

Line 377-382: I am confused by the author's description, why cannot taxa grow? How is a population that cannot grow defined?

Line 384: This sentence should not be a fact and should not be described as 'due to'.

Line 405: What is the basis for the author's statement that the core and CRAT only accounting for a minor fraction of the total diversity in the global AD microbiota? From Figure 5, it can be seen that the number of species in wastewater sludge is the highest, so what is the total global number? Please describe.

Line 441: 'his' should be 'its'?

Line 457-458: Can you determine the configuration of the digester based on the specific sample source?

Line 470: What is knowledge transfer between countries? What knowledge is it about?

In recent years, research on metagenomic-assembly genomes (MAGs) has received increasing attention. By annotating the functions of MAGs and reconstructing metabolic pathways, it is possible to more accurately study the potential functions of taxa. Can the author discuss in conjunction with MAGs in the outlook section to provide better future solutions for the study of microbial communities in AD systems.

Reviewer #2:

Remarks to the Author:

I enjoyed reading the manuscript. It is, indeed, a challenging task to summarize the findings of a large-scale collaboration and showcase the importance and applications of a database.

The authors presented an expansion of the MIDAS 4 database, adding full-length 16S rRNA Amplicon Sequence Variants from bacterial and archaeal species inhabiting Anaerobic Digesters (AD) worldwide. Although this is rather an additive contribution, the authors showed throughout the manuscript the potential applications of the new data. This is the first time some of these taxa have been sequenced, described, and analyzed, which is valuable for the scientific community, and it highlights the importance of reference databases in microbiome studies. The authors suggested a standardized protocol for the collection and amplification of communities from ADs, which, to my point of view, helps minimize batch effect and ensure a fair comparison of communities from different digesters. Community profiling analyses showed that substrate type explains most of the variance across communities. Beyond basic community descriptions, the authors identified core and conditionally rare abundant taxa and provided a thorough explanation and discussion of these results, thereby linking community structure to function. Likewise, the authors identified key organisms associated with

important functions in digesters that may be of special importance for future studies. Overall, this study is an example of large-scale collaborations to bring the hidden prokaryotic biodiversity into light and provide resources to the scientific community.

I have the following questions and suggestions to improve the manuscript:

- Line 168: Mention explicitly how did you obtain the number of 30,246 new FL-ASVs. Did you use a consensus from both databases? This is not clear.
- Line 169: Sequence novelty was determined based on a mapping of the FL-ASVs to SILVA 138.1 SSURef NR99 and MiDAS 4 databases. Both databases have been released recently: 2020 and 2022, respectively. As a suggestion, the authors may also consider analyzing the sequence novelty against the GreenGenes2 database (released in 2022), which is much larger than SILVA, and contains a substantial number of full-length 16S rRNA sequences
- Line 181: Remove the word "highlighted" as there is no reference to a figure or table to show the exact phyla that were introduced.
- Line 182-184: Mention if these lineages have been discarded or kept in the database. If yes, mention the advantages of this new data.
- Line 212: What do you mean by stringent mapping? Mention the "other commonly applied databases" in the main text.
- Line 213-218: The sentence is too long and is hard to keep track of the ideas presented.
- Line 219-220: Can you provide an explanation of why the coverage is not 100% for MiDAS 5? Given that your database is built on reference sequences inferred from the same samples from where the short reads came.
- Line 227-229: Is this statement applicable to MiDAS 5 only or to all databases and classification levels from Figure 2? Mention this explicitly.
- Line 233-233: Indeed MiDAS 5 provides good coverage (except for Archaea as previously stated), but why there is no comparison with the other databases as done before?
- Line 243-244: I find the wording confusing. To what duplicates do you refer?
- Line 259-261: In other words, similar sequences are collapsed to the same reference sequence in MiDAS 4. Still, if your database has a higher resolution (i.e. more reference sequences) you expect it to achieve a greater coverage. Perhaps this is an effect from the parameters used in the mapping step?
- Line 266-267: Great observation! Are these improvements statistically significant?
- Line 318-320: Mention explicitly what metric did you use to derive this claim? (R^2 from Adonis)
- Line 320: add by: "[...]" and to a lesser extent by "[...]"
- Line 328: Fig 4. Fix the orientation of the y-axes labels.
- Line 333-334: Why p-values cannot be confidently determined? You can increase the number of permutations to obtain p-values lower than 0.001
- Line 361: Can you expand on this? How was this achieved?
- Line 368: Change classification with definition to be consistent with the previous paragraph. Otherwise, the reader may think that you redid taxonomic classification.
- Line 395-403: I like this graph. It states the importance of incorporating uncultured taxa in the database. Is it possible to highlight the column/bar that contains the number of shared core species across all samples? This would be easier for the reader.
- Line 420: Do you mean to Supplementary Fig.5?
- Line 441: replace "his" with "this"
- Line 460-462: Is there any reason why you chose to show this subset of countries? Please state your inclusion criteria.
- Line 484-488: This sentence is long and confusing.
- Line 509-511: If I understand correctly, the colors refer to the colored circles. Yet, no yellow or red circles are shown in the figure. Could you clarify this?
- Line 616. Change "taxonomic rank" to "taxonomic ranks"
- Line 682: Mention what UMIs stand for.
- Line 720: Please spell out the full name of "CCS".

Reviewer #3:

Remarks to the Author:

Reviewer #4:

Remarks to the Author:

The present study from Morten Kam Dahl Dueholm and colleagues is focused on anaerobic digestion and the comprehensive sequencing of more than half a million high-quality, full-length 16S rRNA gene sequences from 285 full-scale anaerobic digesters worldwide. This study aimed to expand the understanding of the diversity and function of bacteria and archaea in anaerobic digesters by using amplicon sequencing analysis. The sequences were processed into full-length 16S rRNA amplicon sequence variants, which were integrated into the MiDAS 5 database, enhancing the database coverage. Despite the study is interesting, in my opinion, there are several critical concerns, detailed in the attached feedback. These include methodological limitations, a limited novelty, a very limited comparison with existing literature, and a predominantly descriptive nature of results. Specific comments on individual lines and sections have been provided for your reference.

Methodological Concerns:

Line 62: The use of three sets of primers instead of full-length sequencing requires justification.

Line 683: Clarification is needed regarding whether the amplicons analyzed using PacBio were amplified using 27F and 1391R, considering the potential introduction of amplification bias.

Novelty and Comparison:

Lines 137, 157, and 221-223: The study lacks novelty and robustness in comparison to existing literature. The distribution bias towards European biogas plants (line 137) and the absence of samples from China without reference to existing studies (line 157) limited the contribution of the study. Additionally, the suggestion to compare results with databases (lines 443-445) is not adequately addressed.

Descriptive Nature of Results:

Lines 171-173, 178-179, and 186: The predominantly descriptive results and the discussion of functional roles at the phylum level (line 186) without deeper exploration weaken the study's impact. Additionally, clarity is needed on line 114, and the rationale behind the significant change in the number of sequences from 0.5 million to 120000 (line 154) requires detailed clarification.

References and Databases:

The references section lacks crucial updates related to established databases on the AD microbiome.

Specific Comments:

Line 114: Clarification is needed.

Line 157: The absence of samples from China and failure to reference previous studies on Chinese biogas plants need addressing.

Lines 171-173: 16S rRNA sequences are typically considered as OTUs or ASVs, not for defining assignment to species, genus, etc.

Lines 178-179: A more interesting comparison could be made with global databases rather than a previous version of the same database.

Line 183: Clarity is needed.

Line 221-223: Consideration of samples from independent biogas plants not used in constructing the database is suggested.

Lines 443-445: It is not clear why the results were not compared with already published datasets such as [<https://doi.org/10.1093/gigascience/giaa164>] or [<https://doi.org/10.3389/fmicb.2022.1095497>].

Referee #1 (Remarks to the Author):

This paper reported on the establishment of the MiDAS 5 database, which was based on the sequencing of over 500000 high-quality, full-length 16S rRNA gene sequences from 285 full-scale ADs around the world, obtaining a large amount of data, thereby expanding our understanding of the diversity and function of bacteria and archaea in AD. The author used three sets of common primers targeting different regions of the 16S rRNA gene in bacteria and/or archaea to conduct amplicon based global scale microbial community analysis of sampled AD. It also reveals how environmental conditions and biogeography shape the AD microbiota.

This work has made some exciting new breakthroughs on the basis of their original database, but there are still some minor points that need to be addressed before publication.

Response: Thank you for your positive comments. It is important to note that approximately 1,000,000 of the full-length 16S rRNA gene sequences originate from anaerobic digesters, while the remainder are from wastewater treatment plants (MiDAS 4). Despite this, the data contribute significant novelty to the field.

Specific Comments

12.Line 315-316 I think there should be a reference after this statement.

Response: We agree and have added a reference to the following article: Martiny, J. B. H., Jones, S. E., Lennon, J. T. & Martiny, A. C. Microbiomes in light of traits: A phylogenetic perspective. Science 350, aac9323 (2015).

Line 337-347: This section mainly describes the definitions of certain specific nouns, and it is recommended to reduce or move them up to the introduction section.

Response: We appreciate the suggestion and agree that the section contained some redundant information. We have revised the manuscript by reducing the content as recommended.

Line 377-382: I am confused by the author's description, why cannot taxa grow? How is a population that cannot grow defined?

Response: Thank you for highlighting this point of confusion. By "cannot grow", we intend to indicate that the taxa do not proliferate under the conditions present in the digesters. To eliminate ambiguity, we have revised the phrasing in the manuscript to explicitly state "does not grow in the digesters." in line 396.

Line 384: This sentence should not be a fact and should not be described as 'due to'.

Response: We have changed the heading to "Many core and CRAT represent MiDAS placeholder taxa" in line 404.

Line 405: What is the basis for the author's statement that the core and CRAT only accounting for a minor fraction of the total diversity in the global AD microbiota? From Figure 5, it can be seen that the number of species in wastewater sludge is the highest, so what is the total global number? Please describe.

Response: When we refer to "the global AD microbiota," we specifically mean the ADs that were sampled in our study. To clarify this, we have amended the manuscript to read "the total diversity in the ADs examined." in line 425-427.

We acknowledge the challenge in estimating the total diversity across all global ADs, which is beyond the scope of our current data.

Line 441: 'his' should be 'its'?

Response: We have corrected the text in the revised manuscript.

Line 457-458: Can you determine the configuration of the digester based on the specific sample source?

Response: Thank you for raising this important question. During the revision process, we expanded our amplicon dataset to include results from all three primer sets for each sample. Our updated analysis identified several mesophilic ADs with a high relative abundance of Methanothermobacter. However, we could not establish a definitive link between these observations and specific plant parameters. Consequently, we have revised the relevant paragraphs to reflect this finding:

Line 474-481: "The most common methanogens across substrates and temperatures were Methanoculleus, Methanosarcina, Methanothermobacter, and Methanothermobacter. Methanothermobacter was as expected most abundant in thermophilic ADs. However, to our surprise, it also occurred in high relative abundance in several mesophilic reactors treating mainly food waste. We were not able to explain their occurrences in these ADs based on the available metadata for the plants, but future studies might shed light on the underlying mechanisms or environmental factors that enable this unexpected distribution."

Line 470: What is knowledge transfer between countries? What knowledge is it about?

Response: By "knowledge transfer between countries," we refer to the applicability of findings related to methanogens from one geographic region to others worldwide. Given the high similarity of methanogens across different global locations, it suggests that research insights and best practices developed in one country could potentially be relevant and beneficial in other countries.

See Line 492-494: "The significant similarity of methanogens across various regions indicates substantial potential for global knowledge transfer concerning their management and utilization."

In recent years, research on metagenomic-assembly genomes (MAGs) has received increasing attention. By annotating the functions of MAGs and reconstructing metabolic pathways, it is possible to more accurately study the potential functions of taxa. Can the author discuss in conjunction with MAGs in the outlook section to provide better future solutions for the study of microbial communities in AD systems.

Response: We agree that future studies of MAGs will play a significant role in explaining the role of the observed core and CRAT species. Accordingly, we have added the following statement to the "Conclusion and perspectives":

Line 647-652: "By focusing on the fraction of these abundant and common microbes that can grow in the AD systems, we will be able to explain most of the microbial processes that occur in the anaerobic digestion process. This list of "Most Wanted" organisms contain species that should be prime targets for future in situ studies and the reconstruction of MAGs. These genomes can then be annotated to provide additional details about their potential metabolic pathways and roles in the AD ecosystem^{15,16,76-78}."

Reviewer #2 (Remarks to the Author):

I enjoyed reading the manuscript. It is, indeed, a challenging task to summarize the findings of a large-scale collaboration and showcase the importance and applications of a database.

The authors presented an expansion of the MIDAS 4 database, adding full-length 16S rRNA Amplicon Sequence Variants from bacterial and archaeal species inhabiting Anaerobic Digesters (AD) worldwide. Although this is rather an additive contribution, the authors showed throughout the manuscript the potential applications of the new data. This is the first time some of these taxa have been sequenced, described, and analyzed, which is valuable for the scientific community, and it highlights the importance of reference databases in microbiome studies. The authors suggested a standardized protocol for the collection and amplification of communities from ADs, which, to my point of view, helps minimize batch effect and ensure a fair comparison of communities from different digesters. Community profiling analyses showed that substrate type explains most of the variance across communities. Beyond basic community descriptions, the authors identified core and conditionally rare abundant taxa and provided a thorough explanation and discussion of these results, thereby linking community structure to function. Likewise, the authors identified key organisms associated with important functions in digesters that may be of special importance for future studies. Overall, this study is an example of large-scale collaborations to bring the hidden prokaryotic biodiversity into light and provide resources to the scientific community.

Response: We thank the reviewer for the positive feedback. It's encouraging to hear that the additions to the MIDAS 4 database and our suggested protocols are seen as valuable and motivate us to continue our work and further contribute to the scientific community.

I have the following questions and suggestions to improve the manuscript:

- Line 168: Mention explicitly how did you obtain the number of 30,246 new FL-ASVs. Did you use a consensus from both databases? This is not clear.

Response: In the revised manuscript, we have clarified the method by which we obtained the 30,246 additional FL-ASVs.

Line 152-157: "After processing the sequence reads with AutoTax to produce full-length 16S rRNA gene ASVs (FL-ASVs), these were compared and added to the existing 90,164 FL-ASVs in the MiDAS 4.8.1 database. The combined number was then deduplicated, resulting in a total of 120,408 non-redundant FL-ASV reference sequences in the expanded MiDAS 5 database. This represents an increase of 30,246 new FL-ASVs when compared to the previous version."

- Line 169: Sequence novelty was determined based on a mapping of the FL-ASVs to SILVA 138.1 SSURef NR99 and MiDAS 4 databases. Both databases have been released recently: 2020 and 2022, respectively. As a suggestion, the authors may also consider analyzing the sequence novelty against the GreenGenes2 database (released in 2022), which is much larger than SILVA, and contains a substantial number of full-length 16S rRNA sequences.

Response: In our revised manuscript, we have expanded our database evaluation to include the most recent versions of the Genome Taxonomy Database (GTDB) (both for representative sequences and the complete database) and GreenGenes2 (both the backbone taxonomy and the complete database), see new figure 2, S2, S3, and S4. Our results indicate that SILVA and MiDAS 4 provide superior sequence coverage compared to these additional databases; therefore, we have focused our sequence novelty analysis on these two databases. It is worth noting that although GreenGenes2 contains 21,062,158 sequences, most of these represent V4 ASVs, which are unsuitable for the classification of amplicons from other regions of the 16S rRNA gene. The GreenGenes2 full-length 16S rRNA gene backbone taxonomy only contains 331,269 sequences, which is fewer than those found in SILVA.

- Line 181: Remove the word "highlighted" as there is no reference to a figure or table to show the exact phyla that were introduced.

Response: We have adjusted the sentence as suggested.

- Line 182-184: Mention if these lineages have been discarded or kept in the database. If yes, mention the advantages of this new data.

Response: The sequences representing these lineages have been retained in the database because they represent true biological sequences that can be encountered in the ecosystem. We plan to label them as mitochondrial in a future version of the MiDAS database. By including these sequences, we can use them as decoys to prevent the misclassification of mitochondrial amplicons as bacterial, thus enhancing the accuracy of our database.

- Line 212: What do you mean by stringent mapping? Mention the "other commonly applied databases" in the main text.

Response: By "stringent mapping," we refer to the process conducted without the use of heuristics, which has now been clearly defined in the revised manuscript (see answer to next comment). Regarding the other databases, we have chosen not to mention every applied database in the main text but have noted the inclusion of the recently released GreenGenes2 database. This approach helps to emphasize our focus on using updated and relevant resources for our analysis.

- Line 213-218: The sentence is too long and is hard to keep track of the ideas presented.

Response: Thank you for highlighting this issue. We have revised the text to break down the information into clearer, more manageable parts, as suggested:

Line 218-225: "Our initial analysis involved non-heuristic mapping of short-read ASVs against MiDAS 5 and other widely used reference databases, including the newly released GreenGenes2³⁰. This step allowed us to establish the percent identity between each ASV and its closest match across the databases. We then calculated the percentage of ASVs that have high-identity matches ($\geq 99\%$ identity) in each sample and database. To focus on active microbial populations, we excluded ASVs representing the rare biosphere (those with $< 0.01\%$ relative abundance), which are often enriched in non-growing organisms and environmental DNA^{7,10}."

- Line 219-220: Can you provide an explanation of why the coverage is not 100% for MiDAS 5? Given that your database is built on reference sequences inferred from the same samples from where the short reads came.

Response: There are several factors contributing to the less than 100% coverage in MiDAS 5, despite it is being built from reference sequences derived from the same samples as the short reads. First, there is the issue of primer bias: the primers used to generate the full-length 16S rRNA gene reference sequences differ from those used for the short read amplicons. No primer is perfect, which results in some sequences being detected only in short reads and others only in long reads. Secondly, the sequencing depth plays a significant role. Our short-read sequencing, for example, generated approximately 42.4 million reads for our V1-V3 amplicon dataset, which allows us to resolve more ASVs than is possible with the fewer number of full-length 16S rRNA gene reads available.

- Line 227-229: Is this statement applicable to MiDAS 5 only or to all databases and classification levels from Figure 2? Mention this explicitly.

Response: The statement specifically pertains to MiDAS 5 and its comparison of the percentage of high-identity hits between archaea and bacteria. We have clarified this in the revised manuscript with the following wording:

Line 235-237: "The lower coverage for archaea compared to bacteria in MiDAS 5 is likely due to reduced sequencing efforts and the challenges in designing effective universal primers for archaeal full-length 16S rRNA gene sequencing."

● Line 233-233: Indeed MiDAS 5 provides good coverage (except for Archaea as previously stated), but why there is no comparison with the other databases as done before?

Response: We have previously conducted comprehensive comparisons between MiDAS 5 and other databases, as reported in the earlier sections of our manuscript. Given that these analyses have already established the relative performance and coverage of MiDAS 5, we considered additional direct comparisons to be redundant. Instead, our focus shifted to a more detailed evaluation of MiDAS 5's performance with specific subsets of our data.

● Line 243-244: I find the wording confusing. To what duplicates do you refer?

Response: We appreciate your attention to clarity. To specify, each anaerobic digester in our study was represented by two biological replicates. We have revised the sentence to better reflect this:

Line 255-258: "The violin and box plots illustrate the distribution of the percentage of ASVs with high-identity hits or genus/species-level classifications for each database, analyzed across 570 biologically independent samples, including two biological replicates for each digester."

● Line 259-261: In other words, similar sequences are collapsed to the same reference sequence in MiDAS 4. Still, if your database has a higher resolution (i.e. more reference sequences) you expect it to achieve a greater coverage. Perhaps this is an effect from the parameters used in the mapping step?

Response: Thank you for highlighting this point. It's crucial to clarify that the paragraph discusses classification rather than mapping. Importantly, greater coverage in the database does not necessarily translate to more specific classification. For instance, adding new sequences that closely resemble existing reference sequences in certain variable regions, but differ taxonomically, can complicate the taxonomy assignment. This occurs because the presence of multiple similar sequences might make it more challenging to confidently assign a taxonomy to short reads targeting these regions, a problem that might not exist with fewer reference sequences.

● Line 266-267: Great observation! Are these improvements statistically significant?

Response: Thank you for the positive feedback. We have refined the statement regarding the observed improvements:

Line 280-282: "Interestingly, no statistically significant improvements were observed. This highlights that most of the added references originated from anaerobic digester-specific taxa."

● Line 318-320: Mention explicitly what metric did you use to derive this claim? (R^2 from Adonis)

Response: We have added that this is based on the R^2 value from Adonis in the revised manuscript:

Line 334-337: "The PERMANOVA (Adonis R^2 values) showed that the overall microbial community was mainly explained by the primary substrate and to a lesser extent by temperature, continent, and digester technology (Fig. 4)."

● Line 320: add by: "[...] and to a lesser extent by [...]"

Response: We have rewritten the sentence to improve the readability. See comment above.

- Line 328: Fig 4. Fix the orientation of the y-axes labels.

Response: This has been fixed.

- Line 333-334: Why p-values cannot be confidently determined? You can increase the number of permutations to obtain p-values lower than 0.001

Response: We limited the number of permutations to 999 in our analysis, which inherently sets the lower limit for p-values at 0.001. While it is possible to increase the number of permutations to theoretically obtain lower p-values, practical constraints such as computational resources and time, as well as diminishing statistical returns, often make this approach impractical. We have updated the manuscript to clarify this limitation:

Line 350-351: "Exact p-values less than 0.001 could not be confidently determined due to the chosen number of permutations."

- Line 361: Can you expand on this? How was this achieved?

Response: We have expanded the final sentence so that it explicitly states how the results were combined:

Line 377-379: "To minimize the impact of primer bias, we analyzed all three amplicon datasets and combined the results, including all core and CRAT that were found in at least one of the datasets."

- Line 368: Change classification with definition to be consistent with the previous paragraph. Otherwise, the reader may think that you redid taxonomic classification.

Response: We have revised the paragraph so that it now reads:

Line 386-388: "To define a 'most wanted' list for bacteria and archaea in ADs globally, we linked each core and CRAT to their highest-ranking category across primary substrates, process temperatures, and primer pair (Supplementary Data 3)."

- Line 395-403: I like this graph. It states the importance of incorporating uncultured taxa in the database. Is it possible to highlight the column/bar that contains the number of shared core species across all samples? This would be easier for the reader.

Response: Thank you for the positive feedback on the graph. While it is technically feasible to highlight the specific column that contains the number of shared core species across all samples, we are concerned that doing so might disproportionately emphasize this aspect of the data. Our intent is to present a balanced view of all the data represented in the graph to avoid skewing the focus away from other important findings. We have therefore kept the figure as it is.

- Line 420: Do you mean to Supplementary Fig.5?

Response: Yes, the reference should have been to Supplementary Fig.5 (Supplementary Fig. 6 in the revised manuscript). This has been corrected.

- Line 441: replace "his" with "this"

Response: Done.

- Line 460-462: Is there any reason why you chose to show this subset of countries? Please state your inclusion criteria.

Response: The initial selection of countries in our dataset was inadvertently limited due to errors in our metadata. Upon identifying and correcting these errors, we have updated our dataset to include amplicon data

for all countries and for all primer pairs. This ensures that our analysis now comprehensively reflects the global scope intended for this study.

- Line 484-488: This sentence is long and confusing.

Response: We have broken down and rewritten the sentence to improve clarity:

Line 505-510: "Syntrophic bacteria play a vital role in ADs by converting substrates, such as short-chain fatty acids, into acetate, H₂, and formate. These compounds serve as substrates or reducing equivalents for methanogens, which in turn produce methane and CO₂. This obligately mutualistic metabolism is crucial because the syntrophs can only oxidize substrates and sustain growth under anaerobic conditions if the methanogens rapidly consume the products to maintain them at very low concentrations."

- Line 509-511: If I understand correctly, the colors refer to the colored circles. Yet, no yellow or red circles are shown in the figure. Could you clarify this?

Response: Thank you for your observation. The colors mentioned refer to the status of the syntrophic bacteria: blue circles indicate active growth, while gray circles represent cases where we have no information on their status. The absence of yellow or red circles in the specific figure is because in our dataset, all observed syntrophs are either actively growing (hence depicted in blue) or we lack data about them (depicted in gray).

- Line 616. Change "taxonomic rank" to "taxonomic ranks"

Response: Done.

- Line 682: Mention what UMIs stand for.

Response: Done.

- Line 720: Please spell out the full name of "CCS".

Response: Done.

Reviewer #3 (Remarks to the Author):

Response: We appreciate the collaborative effort in reviewing our manuscript and value the insights provided by both the principal reviewer and the Early Career Researcher involved. We support Nature Communications' initiative to facilitate training in peer review and to recognize the contributions of Early Career Researchers in the peer review process. Thank you for the thorough examination of our work and for the constructive feedback, which has undoubtedly helped enhance the quality of our manuscript.

Reviewer #4 (Remarks to the Author):

The present study from Morten Kam Dahl Dueholm and colleagues is focused on anaerobic digestion and the comprehensive sequencing of more than half a million high-quality, full-length 16S rRNA gene sequences from 285 full-scale anaerobic digesters worldwide. This study aimed to expand the understanding of the diversity and function of bacteria and archaea in anaerobic digesters by using amplicon sequencing analysis. The sequences were processed into full-length 16S rRNA amplicon sequence variants, which were integrated into the MiDAS 5 database, enhancing the database coverage. Despite the study is interesting, in my opinion, there are several critical concerns, detailed in the attached feedback. These include methodological limitations, a limited novelty, a very limited comparison with existing literature, and a predominantly descriptive nature of results. Specific comments on individual lines and sections have been provided for your reference.

Response: We thank Reviewer 4 for their thorough analysis and constructive feedback on our manuscript. We appreciate the time and effort dedicated to reviewing our study, which indeed adds significant value to our work.

We acknowledge the concerns raised regarding methodological limitations, novelty, literature comparison, and the descriptive nature of our results. In response to these concerns, we have carefully revised our manuscript to address each point and have incorporated additional analyses and literature comparisons to enhance the robustness and depth of our findings.

It is worth noting that other reviewers highlighted the strengths of our methodological approach and the utility of the expanded MiDAS 5 database in providing a more comprehensive understanding of microbial communities in anaerobic digesters. These positive comments suggest a balance of perspectives on our study's contributions.

We believe that our revisions, influenced by all reviewers' insights, have strengthened the manuscript significantly. We remain open to further discussion to resolve any outstanding concerns and thank Reviewer 4 again for their valuable contribution to the refinement of our study.

Methodological Concerns:

Line 62: The use of three sets of primers instead of full-length sequencing requires justification.

Response: The selection of three different sets of primers was driven by their common use within the field, which ensures comparability of our results with other studies. Additionally, at the time of conducting this study, high-throughput full-length 16S rRNA sequencing was not economically feasible. Utilizing these primer sets allowed us to balance financial constraints with the need for robust, field-standard data collection, thus maximizing the impact and relevance of our findings within the available resources.

Line 683: Clarification is needed regarding whether the amplicons analyzed using PacBio were amplified using 27F and 1391R, considering the potential introduction of amplification bias.

Response: We believe that this has already been described in the manuscript, see lines 710-713:

“Bacterial and archaeal 16S rRNA genes were UMI-tagged using overhang primers based on the 27F and 1391R⁸⁰ and SSU1ArF and SSU1000ArR³⁴ primer pairs, respectively. These primers have shown excellent coverage for the known bacterial and archaeal diversity in silico^{34,80}.”

While the use of any primer pairs inevitably introduces some degree of amplification bias, the 27F and 1391R primers are known to cover a broad spectrum of bacterial diversity effectively. Furthermore, as the objective of our sequencing was to generating reference sequences, our analysis is less dependent on the relative abundance of the sequences detected. Finally, we have conducted evaluations of our database using short-

read amplicon data generated with different primers to assess coverage of the database and identify any phylogenetic groups that may not be adequately represented.

Novelty and Comparison:

Lines 137, 157, and 221-223: The study lacks novelty and robustness in comparison to existing literature. The distribution bias towards European biogas plants (line 137) and the absence of samples from China without reference to existing studies (line 157) limited the contribution of the study. Additionally, the suggestion to compare results with databases (lines 443-445) is not adequately addressed.

Response: We acknowledge the limitations in geographic distribution, particularly the absence of samples from China and other regions. Despite extensive efforts to include diverse global samples through outreach at conferences and via social media, we encountered challenges in securing cooperation from China and several other countries. To address concerns regarding the representation of unrepresented regions in our study, we have now incorporated external amplicon data from Mei et al. 2019, which includes samples from Canada, Hong Kong, Japan, the Netherlands, and the United States, into our database evaluations. These analyses have demonstrated that the MiDAS 5 database retains global applicability. Moreover, we have broadened our database comparisons to include the latest updates from the genome taxonomy database's 16S rRNA gene sequences and the newly released GreenGenes2 database, further enhancing the robustness and relevance of our study (see also response to reviewer #2 above).

Descriptive Nature of Results:

Lines 171-173, 178-179, and 186: The predominantly descriptive results and the discussion of functional roles at the phylum level (line 186) without deeper exploration weaken the study's impact. Additionally, clarity is needed on line 114, and the rationale behind the significant change in the number of sequences from 0.5 million to 120000 (line 154) requires detailed clarification.

Response: Thank you for your comments. The sections you've highlighted describe the expansion of the MiDAS 16S rRNA gene reference database with full-length 16S rRNA gene ASVs derived from our sequenced anaerobic digesters. This descriptive aspect is intentional and foundational for understanding the database's enhanced diversity. We believe that the descriptive nature of these sections does not diminish the study's impact; rather, it sets the stage for subsequent, more detailed analyses covered later in the article, including the exploration of core and CRAT taxa and the functional roles of key microbial guilds.

For line 114, we have improved the text to enhance clarity:

Line 109-115: "AutoTax provides a comprehensive seven-rank taxonomy (kingdom to species-level) for all reference sequences based on the latest SILVA SSURef 99 NR taxonomy and includes a robust placeholder taxonomy for lineages lacking an official classification. These placeholders are named 'midas_x_y', where 'x' represents the taxonomic rank and 'y' a numerical identifier, facilitating comparative studies across various taxonomic ranks."

Regarding the query about sequence numbers (line 154), the initial count of 0.5 million referred to sequence reads, which includes redundant sequences. We've clarified this by specifying 'sequence reads' in the revised manuscript to better reflect the distinction between raw reads and unique ASVs, which are fewer in number. In addition, we have elaborated on how the new ASVs were merged with the previous version of the MiDAS database:

Line 152-157: "After processing the sequence reads with AutoTax to produce full-length 16S rRNA gene ASVs (FL-ASVs), these were compared and added to the existing 90,164 FL-ASVs in the MiDAS 4.8.1 database. The combined number was then deduplicated, resulting in a total of 120,408 non-redundant FL-ASV reference

sequences in the expanded MiDAS 5 database. This represents an increase of 30,246 new FL-ASVs when compared to the previous version.”

References and Databases:

The references section lacks crucial updates related to established databases on the AD microbiome.

Response: We appreciate your comment regarding the inclusion of established databases on the anaerobic digester (AD) microbiome and have included references to the articles specified by you in your comments below. However, as these databases created in the studies are not 16S rRNA gene reference databases they cannot be directly apply for our 16S rRNA gene amplicon data sets.

Specific Comments:

Line 114: Clarification is needed.

Response: We have rewritten this paragraph as stated above.

Line 157: The absence of samples from China and failure to reference previous studies on Chinese biogas plants need addressing.

Response: We acknowledge the importance of including samples from China given its significant role in biogas production. Despite our extensive efforts to engage collaborators through personal contacts, conferences, and social media, we were unable to secure a partnership in China and several other countries. We understand the value of comprehensive geographic representation and regret any gaps in our dataset. We strive for a balanced global survey and do not prioritize one country over others when faced with sampling limitations. We are committed to expanding our collaborations in future studies to include a more diverse range of samples.

Lines 171-173: 16S rRNA sequences are typically considered as OTUs or ASVs, not for defining assignment to species, genus, etc.

Response: Thank you for your comment. You are correct in noting that 16S rRNA gene sequences have traditionally been used to define operational taxonomic units (OTUs) or amplicon sequence variants (ASVs), which are proxies for microbial taxa but not necessarily precise indicators of specific taxonomic levels like species or genus. While OTUs often serve as species-level proxies based on sequence similarity thresholds, ASVs are exact sequences that can provide finer resolution without predefined taxonomic assignment. To estimate taxonomic diversity at various levels from our sequences, we use the established statistical thresholds proposed by Yarza et al. These thresholds, while useful, are average estimates and do not uniformly apply across the bacterial phylogenetic tree. We acknowledge this limitation and emphasize that our taxonomic assignments are approximations intended to facilitate biological interpretation in the revised manuscript:

Line 174-176: “It should be noted that these thresholds do not uniformly apply across the bacterial phylogenetic tree; therefore, our taxonomic assignments should be considered as approximations intended to facilitate biological interpretation.”

Lines 178-179: A more interesting comparison could be made with global databases rather than a previous version of the same database.

Response: We appreciate your suggestion to broaden the scope of our database comparisons. In response, we have expanded our analysis in the revised manuscript to include comparisons with universal reference databases such as the complete 16S rRNA gene database from the Genome Taxonomy Database (GTDB) and the recently released GreenGenes2 database. These additional comparisons provide further evidence

supporting the robustness and utility of MiDAS 5 over other databases. (see also response to reviewer # 2 above).

Line 183: Clarity is needed.

Response: We have revised the paragraph to improve clarity:

Line 188-190: “In addition, we identified nine lineages classified as MiDAS placeholder phyla. However, phylogenetic analysis revealed that these lineages branch closely to mitochondrial sequences, indicating they are likely mitochondrial in origin.”

Line 221-223: Consideration of samples from independent biogas plants not used in constructing the database is suggested.

Response: We appreciate your suggestion and have responded by incorporating an evaluation of our database using amplicon data from Mei et al. 2017 in the revised manuscript (new supplementary fig. 3). This dataset includes samples from 90 anaerobic digesters that were not involved in the construction of our database. Our analysis demonstrated that MiDAS 5 performs robustly also on samples unrelated to our study, validating its applicability and effectiveness.

Reference: Mei, R. et al. Operation-driven heterogeneity and overlooked feed-associated populations in global anaerobic digester microbiome. Water Research 124, 77–84 (2017).

Lines 443-445: It is not clear why the results were not compared with already published datasets such as [<https://doi.org/10.1093/gigascience/giaa164>] or [<https://doi.org/10.3389/fmicb.2022.1095497>].

Response: We thank the reviewer for reminding us of the two excellent papers, which we have referenced in our revised manuscript to support the importance of large-scale studies and MAGs.

Reviewers' Comments:

Reviewer #1:

Remarks to the Author:

Thank you for a nice, rigorous revision of your manuscript. I have no further comments.

Reviewer #2:

Remarks to the Author:

We have reviewed the revised manuscript. Overall, the authors have responded to all our questions and clarified statements when needed. They have also performed benchmarks with two additional databases, thus highlighting MIDAS 5's good performance. We also thank the authors for analyzing an independent amplicon dataset, which provides a fairer performance measure for the expanded database. Altogether, the authors have improved the quality of the manuscript. We, therefore, endorse the current version for publication.

Reviewer #3:

Remarks to the Author:
